# FXR Signaling-Mediated Bile Acid Metabolism Is Critical for Alleviation of Cholesterol Gallstones by *Lactobacillus* Strains

Xin Ye,[a,b] Dan Huang,[a,b] Zhixia Dong,[a,b] Xiaoxin Wang,[c] Min Ning,[a,b] Jie Xia,[a,b] Shuang Shen,[a,b] Shan Wu,[a,b] Yan Shi,[a,b] ⓘ Jingjing Wang,[c] ⓘ Xinjian Wan[a,b]

[a]Shanghai Jiao Tong University School of Medicine Affiliated Sixth People's Hospital, Shanghai, China
[b]Digestive Endoscopic Center, Shanghai Jiao Tong University Affiliated Sixth People's Hospital, Shanghai, China
[c]Shanghai Key Laboratory of Pancreatic Diseases, Institute of Translational Medicine, Shanghai General Hospital, Shanghai Jiao Tong University School of Medicine, Shanghai, China

Xin Ye and Dan Huang contributed equally to this work. Author order was determined on the basis of seniority.
Xinjian Wan and Jingjing Wang shared senior authorship.

**ABSTRACT** Cholesterol gallstone (CGS) disease is characterized by an imbalance in bile acid (BA) metabolism and is closely associated with gut microbiota disorders. However, the role and mechanism by which probiotics targeting the gut microbiota attenuate cholesterol gallstones are still unknown. In this study, *Limosilactobacillus reuteri* strain CGMCC 17942 and *Lactiplantibacillus plantarum* strain CGMCC 14407 were individually administered to lithogenic-diet (LD)-fed mice for 8 weeks. Both *Lactobacillus* strains significantly reduced LD-induced gallstones, hepatic steatosis, and hyperlipidemia. These strains modulated BA profiles in the serum and liver, which may be responsible for the activation of farnesoid X receptor (FXR). At the molecular level, *L. reuteri* and *L. plantarum* increased ileal fibroblast growth factor 15 (FGF15) and hepatic fibroblast growth factor receptor 4 (FGFR4) and small heterodimer partner (SHP). Subsequently, hepatic cholesterol 7α-hydroxylase (CYP7A1) and oxysterol 7α-hydroxylase (CYP7B1) were inhibited. Moreover, the two strains enhanced BA transport by increasing the levels of hepatic multidrug resistance-associated protein homologs 3 and 4 (Mrp3/4), hepatic multidrug resistance protein 2 (Mdr2), and the bile salt export pump (BSEP). In addition, both *L. reuteri* and *L. plantarum* reduced LD-associated gut microbiota dysbiosis. *L. reuteri* increased the relative abundance of *Muribaculaceae*, while *L. plantarum* increased that of *Akkermansia*. The changed gut microbiota was significantly negatively correlated with the incidence of cholesterol gallstones and the FXR-antagonistic BAs in the liver and serum and with the FXR signaling pathways. Furthermore, the protective effects of the two strains were abolished by both global and intestine-specific FXR antagonists. These findings suggest that *Lactobacillus* might relieve CGS through the FXR signaling pathways.

**IMPORTANCE** Cholesterol gallstone (CGS) disease is prevalent worldwide. None of the medical options for prevention and treatment of CGS disease are recommended, and surgical management has a high rate of recurrence. It has been reported that the factors involved in metabolic syndrome are highly connected with CGS formation. While remodeling of dysbiosis of the gut microbiome during improvement of metabolic syndrome has been well studied, less is known about prevention of CGS formation after regulating the gut microbiome. We used the lithogenic diet (LD) to induce an experimental CGS model in C57BL/6J mice to investigate protection against CGS formation by *Limosilactobacillus reuteri* strain CGMCC 17942 and *Lactiplantibacillus plantarum* strain CGMCC 14407. We found that these *L. reuteri* and *L. plantarum* strains altered the bile acid composition in mice and improved the dysbiosis of the gut microbiome. These two *Lactobacillus* strains prevented CGS formation by fully activating the

Address correspondence to Xinjian Wan, slwanxinjian2020@126.com, or Jingjing Wang, wangjingjing6891@163.com.

The authors declare no conflict of interest.

hepatic and ileal FXR signaling pathways. They could be a promising therapeutic strategy for treating CGS or preventing its recurrence.

**KEYWORDS** cholesterol gallstones, *Lactobacillus*, FXR, FGF15, bile acid, gut microbiota

The incidence rate of the formation of cholesterol gallstones (CGS) is increasing with the popularity of Western food (1, 2). CGS are caused by cholesterol precipitation in the gallbladder or biliary tract, which can lead to more serious diseases, such as cholecystitis, cholangitis, pancreatitis, gallbladder carcinoma, and colon cancers (3, 4). Ursodeoxycholic acid (UDCA) has been used for dissolving CGS in select patients but fell out of favor because of the high rates of recurrent CGS formation (5). Currently, none of the medical options for the prevention and treatment of CGS are recommended, and surgical management has a high rate of recurrence (2). Therefore, it is necessary to find effective therapies to both prevent and treat CGS.

Gut microbiota dysbiosis and bacterial community assembly were associated with cholesterol gallstones in a large-scale study (6–8). A new study found that fecal transplantation of gut microbiota from gallstone patients to a gallstone-resistant strain of mice could induce gallstone formation (9). It has been reported that the proportion of *Lactobacillus* is greatly reduced in CGS mouse models (10); however, whether *Lactobacillus* supplementation can reduce gallstones has not been studied. Furthermore, Takeda and colleagues showed that *Clostridium butyricum* supplementation could alleviate the formation of gallstones in experimental CGS (11), but they did not state the mechanism by which the studied strain alleviated gallstones. Furthermore, it has been reported that *Limosilactobacillus reuteri* strain CBG-C15 and *Lactiplantibacillus plantarum* strain CBG-C21 had beneficial effects against hypercholesterolemia in rats (12). Recently, research has shown that *Limosilactobacillus reuteri* strain Fn041 prevented hypercholesterolemia by promoting cholesterol and bile acid (BA) excretion in mice (13). Liang and colleagues showed that *Lactiplantibacillus plantarum* strain H-87 had cholesterol-lowering potential due to its role in regulating BA metabolism (14). These results suggested that bacterial strains belonging to these two *Lactobacillus* species might have the potential to regulate cholesterol-related metabolism. We previously isolated *Limosilactobacillus reuteri* strain CGMCC 17942 and *Lactiplantibacillus plantarum* strain CGMCC 14407 (*L. reuteri* and *L. plantarum* hereinafter) from the feces of healthy adults with normal serum cholesterol levels, and we conducted this study to determine whether supplementation with these two strains could reduce gallstone formation and identify the mechanism.

BAs are biosynthesized and conjugated in hepatocytes and then secreted through the gallbladder into the small intestine, where they are hydrolyzed and dehydroxylated by the gut microbiota and a fraction are reabsorbed into the liver through the portal system in the intestine (15). Alteration of the content or composition of BAs leads to the precipitation of cholesterol in bile, which is considered to be an important cause of gallstones (16). The gut microbiota-mediated transformation of the BA pool regulates the activation of nuclear receptor farnesoid X receptor (FXR) pathway signaling (15, 17, 18). FXR, as a member of the steroid/thyroid hormone receptor family of ligand-activated transcription factors, is an important BA receptor governing bile acid, glucose, and lipid metabolism (15, 19). The current research indicates that FXR protects against CGS formation by regulating the hydrophobicity of BAs to promote the excretion of cholesterol (20, 21), inhibit the biosynthesis of BAs and cholesterol, and reduce cholesterol crystal precipitation (15, 19). Moschetta and colleagues showed that FXR activation could increase the expression of the small heterodimer partner (SHP) gene to regulate BA anabolism and increase the transport of biliary phospholipids and bile salts to promote cholesterol solubilization in bile (20).

Accumulating evidence in other disease models (not CGS) indicates that oral supplementation with probiotics, such as *Lactobacillus rhamnosus* strain GG, *Lactobacillus casei* strain YRL577, and probiotic mixture VSL#3, can suppress hepatic BA synthesis and enhance BA deconjugation or excretion through the FXR-fibroblast growth factor 15 (FGF15) signaling pathway in mice (17, 22, 23). However, whether the effects of

probiotics on CGS depend on the FXR-FGF15 axis is unknown. Therefore, we conducted this study to determine whether supplementation of *L. reuteri* or *L. plantarum* would reduce CGS formation by activating the FXR signaling pathways.

In this study, our data showed that supplementation with *L. reuteri* and *L. plantarum* effectively prevented CGS formation. These strains ameliorated gut microbiota dysbiosis by increasing the relative abundance of *Muribaculaceae* or *Akkermansia*. *L. reuteri* and *L. plantarum* treatments maintained the homeostasis of BA metabolism, primarily by reducing tauro-$\alpha$-muricholic acid (T-$\alpha$-MCA) and tauro-$\beta$-MCA (T-$\beta$-MCA) to increase FXR activity and reduce BA and cholesterol synthesis in the liver. Both global and intestine-specific FXR antagonists abolished the protective effects of the two strains on CGS.

## RESULTS

***L. reuteri* and *L. plantarum* treatments reduced LD-induced gallstones and metabolic disorders in mice.** We used a lithogenic-diet (LD)-induced CGS model in mice to investigate whether *L. reuteri* and *L. plantarum* could prevent the formation of CGS. Mice were fed LD for 8 weeks, and *L. reuteri* and *L. plantarum* were supplemented once a day from the beginning of the experiment. The gallbladders of mice fed a normal diet (ND) were filled with clear bile and were without any cholesterol crystals. Round gallstones or stratified crystals and a small amount of sticky bile occupied the gallbladders of mice fed the LD. The incidence of gallstones reached 100%. Only a few leaflet crystals or some cholesterol particles that were invisible to the naked eye and viscous bile were observed in the gallbladders of mice with *L. reuteri* and *L. plantarum* treatments, and the incidence of gallstones was markedly reduced (Fig. 1A and B). We used the grading standards developed by Akiyoshi and colleagues to evaluate the gallstones (24). We found that the grade of the experimental gallstones in mice fed the LD was significantly higher than that of mice fed the ND and was decreased noticeably with *L. reuteri* and *L. plantarum* treatments (Fig. 1C). Next, we evaluated the lipid profiles in gallbladder bile and calculated the cholesterol saturation index (CSI) to understand the mechanism involved in the reduction of CG formation by *L. reuteri* and *L. plantarum*. Compared with that in the ND group, the bile total cholesterol (TC) level was increased significantly in mice fed the LD, whereas the *L. reuteri* and *L. plantarum* treatments decreased the TC level significantly. Compared with that in the ND group, the level of phospholipids (PL) was significantly decreased in the bile of LD mice, whereas the *L. reuteri* and *L. plantarum* treatments increased the PL level significantly. However, the level of total bile acids (BAs) in bile was increased significantly and did not alter significantly with *L. reuteri* or *L. plantarum* treatment (Fig. 1D). The CSI in the LD group was higher than that in the ND group, and *L. reuteri* and *L. plantarum* treatments both reduced it (Fig. 1E), suggesting that *L. reuteri* and *L. plantarum* treatments could improve the LD-induced cholesterol hypersaturation in bile.

Several lines of evidence suggest that metabolic syndrome (MS) is associated with the morbidity of gallstones (25, 26). We monitored the weights of mice once a week and found that the weights of LD-fed mice were significantly increased compared with those of ND-fed mice. The weight gains were inhibited in mice receiving *L. reuteri* and *L. plantarum* treatments (Fig. 1F). We next calculated the ratio of liver weight to body weight in mice. Compared with ND-fed mice, the ratio of liver weight to body weight was significantly higher in LD-fed mice, and *L. reuteri* treatment could reverse the change significantly. The ratio in *L. plantarum*-treated mice did not decrease compared with the ratio in LD-fed mice and was significantly different from that of the ND group (Fig. 1G). The significant increase of the ratio in the *L. plantarum*-treated group was likely because of the body weight loss after *L. plantarum* treatment (Fig. 1G). It was reported that insulin resistance was closely related to the onset of gallstones (26, 27). Therefore, we further examined the serum glucose and insulin functions of mice by the oral glucose tolerance test (OGTT). We found that the hyperglycemia of LD-fed mice was significantly improved by *L. reuteri* and *L. plantarum* treatments. Furthermore, the glucose tolerance of LD-fed mice was severely impaired compared to that in ND-fed

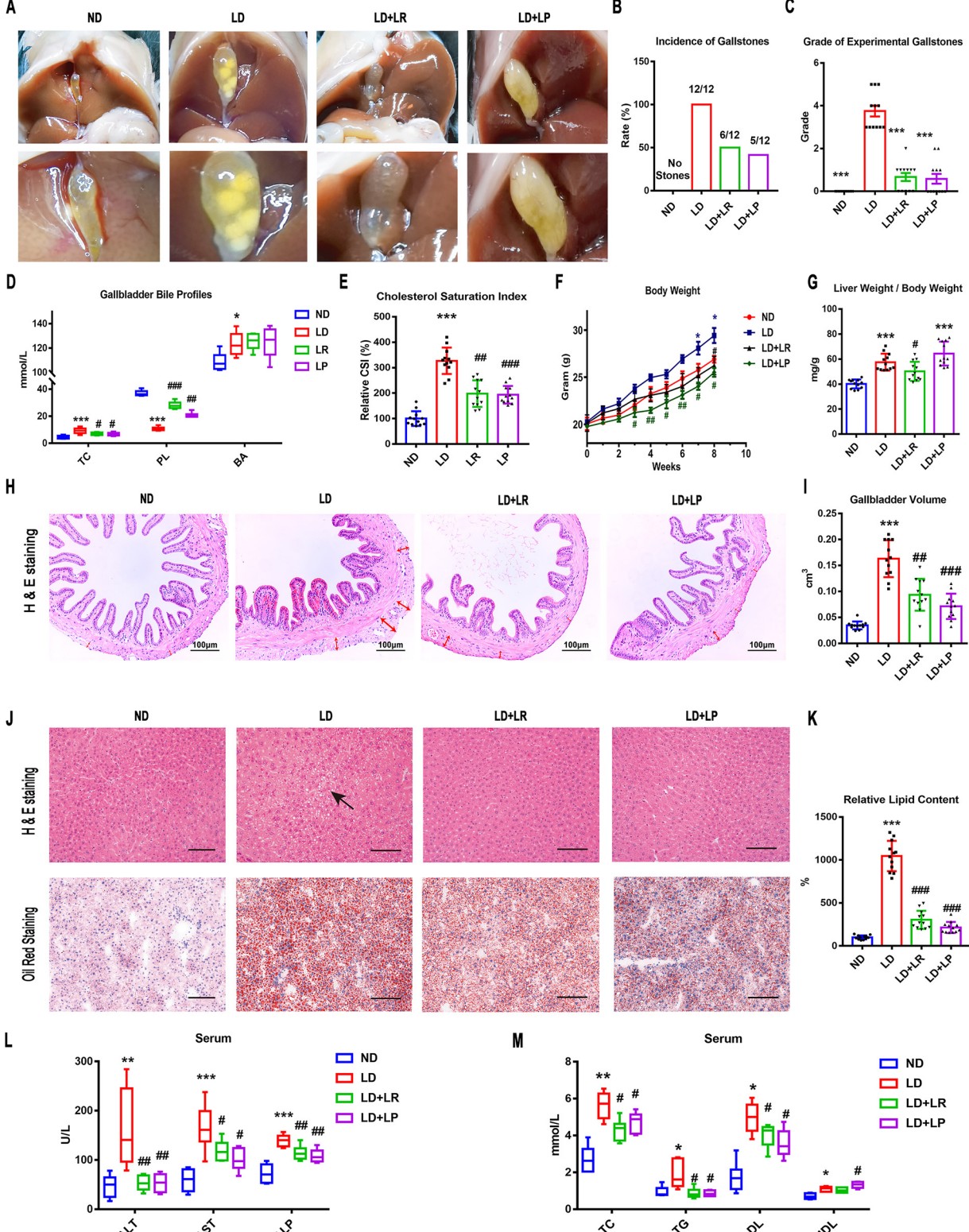

**FIG 1** *L. reuteri* (LR) and *L. plantarum* (LP) treatments reduced LD-induced gallstones and metabolic disorders in mice. Twelve mice were randomly assigned to each group and fed a normal diet (ND) or a lithogenic diet (LD) with or without *L. reuteri* or *L. plantarum* treatment (10⁹ CFU/day) for 8 weeks. (A) Gross appearance of gallbladders and gallstones of mice administered different treatments. (B) Percentage of gallstone incidence in each group of mice. (C) The grades of experimental cholesterol gallstones (CGSs) in the mice were based on the observed cholelithiasis. (D) Total cholesterol (TC), bile acids (BA), and phospholipids (PL) in gallbladders. (E) CSI in each group of mice. (F) Body weights of mice were recorded once a week. (G) Ratios of liver weight to body weight. (H) Representative images of H&E-stained

mice. It was effectively improved by *L. reuteri* and *L. plantarum* treatments (Fig. S1A and B in the supplemental material). These results suggested that the improvements of lipid metabolism and insulin resistance were accompanied by the reduction of CGS formation by *L. reuteri* and *L. plantarum* treatments.

LD administration resulted in significant histopathological and biochemical changes in mice. We found that the connective tissue of the gallbladder's tunica adventitia was thicker and looser in LD-fed mice than in the ND group, and these alterations were reduced by *L. reuteri* and *L. plantarum* treatments (Fig. 1H). Compared with that in the ND group, the gallbladder volume increased significantly in the LD-fed mice, and the change was markedly reversed by *L. reuteri* and *L. plantarum* treatments (Fig. 1I). These alterations of the gallbladder suggested that *L. reuteri* and *L. plantarum* treatments would be involved in CGS prevention by improving the hypertrophy of the gallbladder wall and the physiological function of the gallbladder. Next, we evaluated the improvement of hepatic steatosis and metabolite abnormalities with *L. reuteri* and *L. plantarum* supplementations. Compared with the liver parenchyma in the ND group, there were many vacuoles in the liver parenchyma of LD-fed mice, led by the accumulation of lipid droplets. *L. reuteri* and *L. plantarum* treatments led to a marked reduction in liver histological damage (Fig. 1J). Oil red O staining was used to further confirm the change of lipids in the liver parenchyma. Compared with that in the ND group, the liver parenchyma of LD-fed mice showed many lipid droplets stained with oil red. The quantity of lipid droplets was significantly decreased by *L. reuteri* and *L. plantarum* treatments (Fig. 1J and K). Furthermore, compared with the levels in the ND group, the levels of serum alanine aminotransferase (ALT), aspartate aminotransferase (AST), alkaline phosphatase (ALP), total cholesterol (TC), and triglycerides (TG) were increased in the LD group, whereas *L. reuteri* and *L. plantarum* treatments decreased them significantly (Fig. 1L and M). The levels of high-density lipoprotein (HDL) and low-density lipoprotein (LDL) were increased in the LD group compared with the levels in the ND group. The level of LDL was decreased with *L. reuteri* and *L. plantarum* treatments. The level of HDL was further increased in *L. plantarum*-treated mice (Fig. 1M). These data suggested that the oral administration of *L. reuteri* and *L. plantarum* could ameliorate the histological damage in the gallbladder and liver and improve the hepatic steatosis and liver lipid metabolism, which is associated with reduction of CGS formation (28, 29).

**_L. reuteri_ and _L. plantarum_ treatments altered the bile acid composition in liver and serum in LD-fed mice.** Bile acids (BAs), as important signaling molecules, participate in the regulation of BA and cholesterol metabolism (30). Most BAs are reabsorbed from the intestinal epithelium after being processed by gut microbes and then entering the enterohepatic circulation (31, 32). In the current study, we measured the total bile acids and the composition of BAs in mouse liver and serum to understand the mechanism involved in the reduction of CGS formation by *L. reuteri* and *L. plantarum* treatments. We found that the level of total liver BAs was significantly increased in LD-fed mice compared with that in ND-fed mice and did not change with *L. reuteri* and *L. plantarum* treatments (Fig. 2A). The level of total serum BAs was markedly increased in LD-fed mice compared with the level in ND-fed mice and was significantly decreased by oral *L. reuteri* treatment (Fig. 2B). BA-targeted metabolomics analysis was employed to determine the alterations in individual BAs in the liver and serum of mice. We found that the compositions of BAs in the liver and serum of LD-fed mice were significantly different from those in ND-fed mice (Fig. 2C and D). We analyzed the top 12 BAs in the serum and liver, respectively, and found that the levels of cholic acid (CA) and tauro-

**FIG 1** Legend (Continued)
gallbladder sections (×200). The tunica adventitia of the gallbladder is indicated by red arrows. (I) Gallbladder volumes were estimated by the length, diameter, and circumference of the gallbladders. (J) Representative images of H&E-stained and oil red O-stained liver sections (×200). (K) Percentages of oil red O-positive areas. (L and M) Serum alanine aminotransferase (ALT), aspartate aminotransferase (AST), alkaline phosphatase (ALP), total cholesterol (TC), triglyceride (TG), high-density lipoprotein (HDL), and low-density lipoprotein (LDL) were determined by using a Siemens fast automatic biochemical analyzer (Advia 2400). Data were analyzed by ANOVA along with the *post hoc* Tukey test and are presented as the mean values ± SEM ($n = 12$). *, $0.01 < P \leq 0.05$; **, $0.001 < P \leq 0.01$; ***, $P \leq 0.001$ versus ND group; #, $0.01 < P \leq 0.05$; ##, $0.001 < P \leq 0.01$; ###, $P \leq 0.001$ versus LD group.

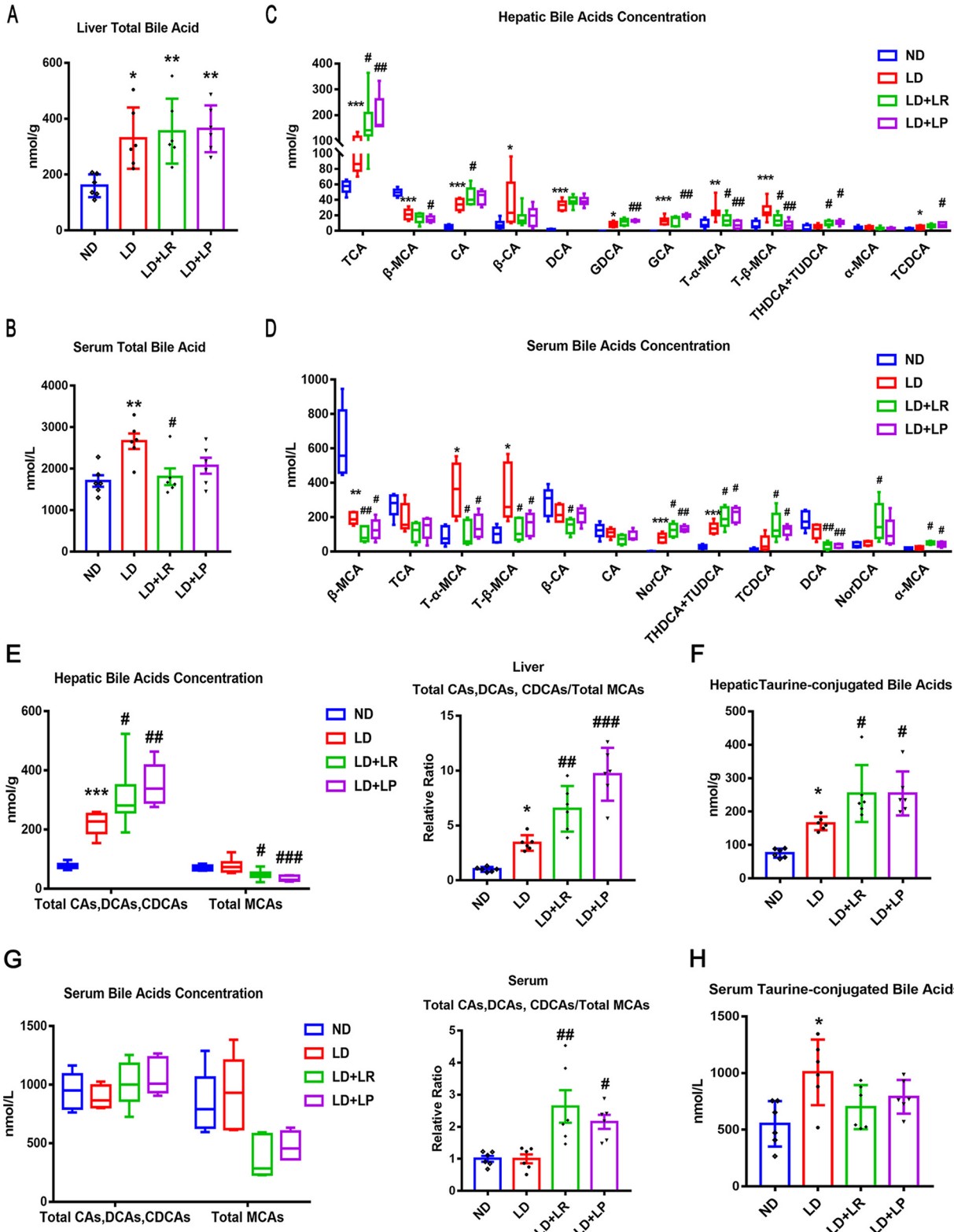

**FIG 2** *L. reuteri* (LR) and *L. plantarum* (LP) treatments changed the bile acid species in the liver and serum. The mice in each group were sacrificed at the end of the 8th week, and serum and liver were collected. (A) Total bile acids (TBA) in the livers of mice. (B) TBA in the serum of mice. (C) The top 12 most abundant hepatic BA species were analyzed. (D) The top 12 most abundant serum BA species are shown. NorCA, norcholic acid; NorDCA, 23-nordeoxycholic acid. (E) Total CAs, DCAs, and CDCAs, total MCAs, and ratios of total CAs, DCAs, and CDCAs to total MCAs in the livers. (F) Total taurine-conjugated BAs in the livers. (G) Total CAs, DCAs, and CDCAs, total MCAs, and ratios of total CAs, DCAs, and CDCAs to total MCAs in the serum. (H) Total taurine-conjugated BAs in the serum. Data were analyzed by ANOVA along with the *post hoc* Tukey test and are presented as the mean values ± SEM (*n* = 6). *, $0.01 < P \leq 0.05$; **, $0.001 < P \leq 0.01$; ***, $P \leq 0.001$ versus normal diet (ND) group; #, $0.01 < P \leq 0.05$; ##, $0.001 < P \leq 0.01$; ###, $P \leq 0.001$ versus lithogenic diet (LD) group.

cholic acid (TCA) in the liver were significantly increased in the LD group compared with their levels in the ND group and that there were more marked increases in CA and TCA with *L. reuteri* and *L. plantarum* treatments (Fig. 2C). However, the levels of CA and TCA did not increase in the serum of LD-fed mice or *L. reuteri*- or *L. plantarum*-treated mice (Fig. 2D). In addition, we found that taurochenodeoxycholic acid (TCDCA), the most potent endogenous FXR agonist, with a 50% effective concentration (EC$_{50}$) of 17 $\mu$M (33), was increased in the liver and serum of LD-fed mice and that *L. reuteri* and *L. plantarum* treatments further upregulated the increased TCDCA level induced by LD feeding. There was a study that demonstrated that $\beta$-muricholic acid ($\beta$-MCA), a major bile acid in mice, functions as an antagonist of the FXR (34). We found that the levels of $\beta$-MCA in the liver and serum were significantly decreased with LD treatment, and there were more marked decreases in $\beta$-MCA with *L. reuteri* and *L. plantarum* treatments (Fig. 2C and D). Furthermore, tauro-$\alpha$-muricholic acid (T-$\alpha$-MCA) (50% inhibitory concentration [IC$_{50}$] = 28 $\mu$M) and T-$\beta$-MCA (IC$_{50}$ = 40 $\mu$M) are both known as efficient natural antagonists of FXR (34). We found that the levels of T-$\alpha$-MCA and T-$\beta$-MCA were significantly increased in the LD group compared with the levels in the ND group and that their levels were markedly decreased by *L. reuteri* and *L. plantarum* treatments (Fig. 2C and D). It has been reported that downregulation of T-$\alpha$-MCA and T-$\beta$-MCA can reverse the inhibition of the FXR and activate the FXR (33, 35). To further demonstrate that the FXR was activated by *L. reuteri* and *L. plantarum* treatments, the ratio of total cholic acids (CAs), deoxycholic acids (DCAs), and chenodeoxycholic acids (CDCAs) (containing taurine-conjugated BAs), which were taken as FXR-agonistic BAs, to total MCAs, which were taken as FXR-antagonistic BAs, was calculated. Compared with that in ND-fed mice, the ratio mentioned above was significantly higher in LD-fed mice, and *L. reuteri* and *L. plantarum* treatments could further increase the ratio of hepatic BAs (Fig. 2E). In addition, the ratio of serum BAs did not increase in the LD group compared with that in the ND group; however, the ratio was markedly increased with *L. reuteri* and *L. plantarum* treatments (Fig. 2G). These results suggest that FXR might have been activated in LD-fed mice and that the activity of FXR was further enhanced by *L. reuteri* and *L. plantarum* treatments. It has been reported that taurine-conjugated BAs have higher hydrophilicity and a higher affinity for FXR, which is beneficial to prevent the formation of CGS (33). The total taurine-conjugated BAs, including TCA, TCDCA, tauroursodeoxycholic acid (TUDCA), taurohyodesoxycholic acid (THDCA), T-$\alpha$-MCA, and T-$\beta$-MCA, were significantly upregulated in the liver and serum of LD-fed mice compared with those of ND-fed mice, and there were more marked increases of the hepatic taurine-conjugated BAs with *L. reuteri* and *L. plantarum* treatments (Fig. 2F and H). These results suggest that the *L. reuteri* and *L. plantarum* treatments might reduce the formation of CGS by increasing the FXR-agonistic BAs, decreasing the FXR-antagonistic BAs, and upregulating the hydrophilicity of BAs.

**_L. reuteri_ and _L. plantarum_ treatments activated the FXR–FGF15/SHP signaling pathway.** FXR and the associated signaling molecules were detected to further determine whether FXR signaling was activated in the ileum and liver. We examined the mRNA and protein levels of FGF15 in the ileum and found that the FGF15 expression in the ileum was significantly increased in LD-fed mice compared with the level in ND-fed mice. Furthermore, there were more marked increases of the mRNA and protein levels of FGF15 with *L. reuteri* and *L. plantarum* treatments, and the expression of FGF15 in the ileum detected by immunofluorescence was consistent with the mRNA and protein levels (Fig. 3A to C). There were no significant differences in the mRNA and protein levels of FXR in the ileum and liver between the ND and LD groups, and there were still no significant changes after *L. reuteri* and *L. plantarum* treatments (Fig. S2A to D).

Furthermore, the mRNA level of liver *Shp* was significantly increased in LD-fed mice and further increased by *L. reuteri* and *L. plantarum* treatments (Fig. 3D). It is worth noting that the mRNA and protein levels of liver fibroblast growth factor receptor 4 (FGFR4), an important receptor in the FXR signaling pathway, were found to be markedly reduced in LD-fed mice compared with the levels in ND-fed mice and that the levels were reversed by *L. reuteri* and *L. plantarum* treatments (Fig. 3E and F). These results

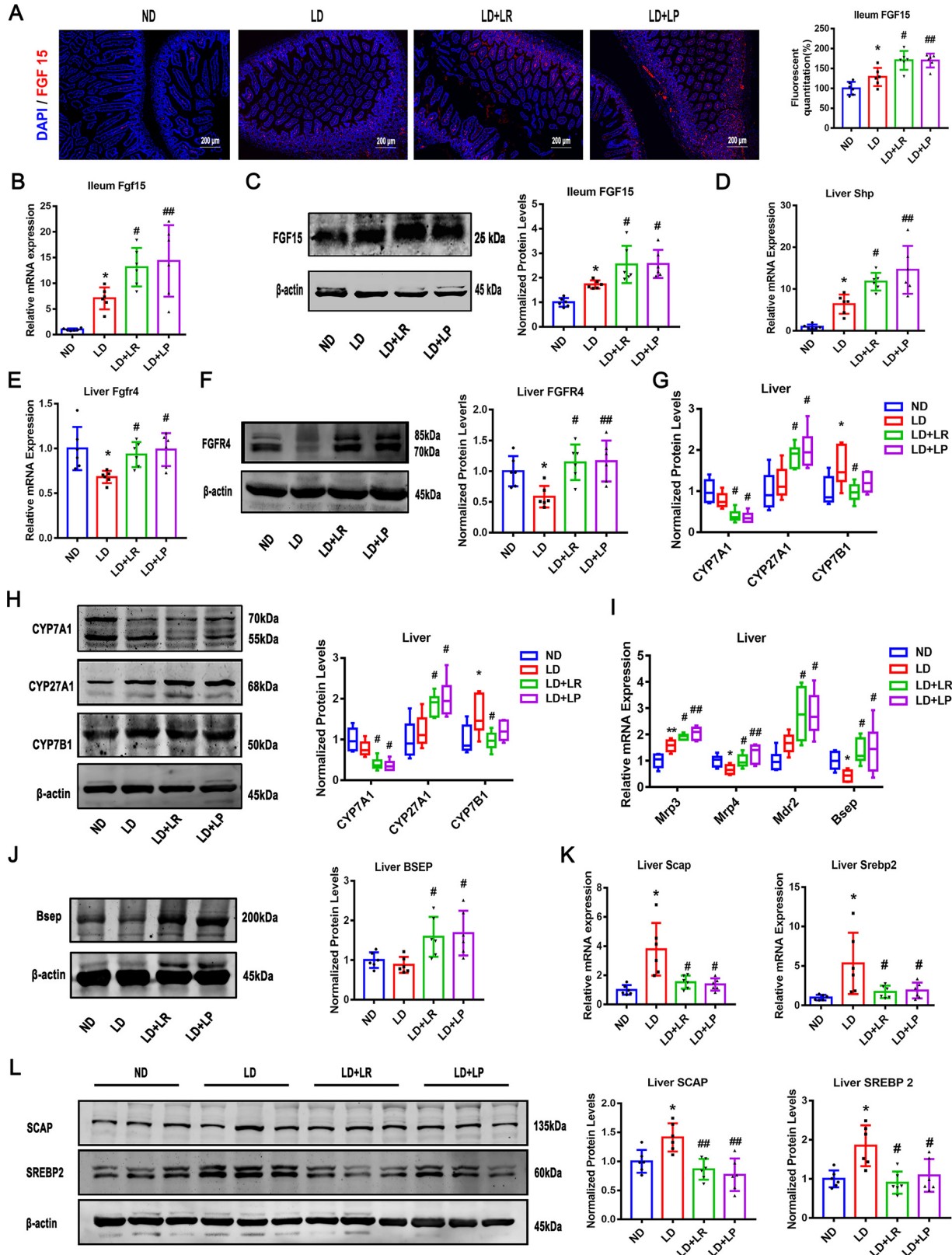

**FIG 3** *L. reuteri* (LR) and *L. plantarum* (LP) treatments activated FXR signaling pathway. The mice in each group were sacrificed at the end of the 8th week, and serum and ileum samples were collected. (A) Immunofluorescence analyses of ileal FGF15 (×400). (B) Ileal mRNA expression of *Fgf15*. (C) Protein expression and quantification of ileal FGF15. (D and E) Hepatic mRNA levels of *Shp* and Fgfr4. (F) Protein expression and quantification of hepatic FGFR4. (G and H) Hepatic mRNA and protein expression of CYP7A1, CYP27A1, and CYP7B1 and

indicated that the FXR-FGF15-FGFR4 signaling pathway could be impaired with LD treatment and that *L. reuteri* and *L. plantarum* administration could repair it. To study the downstream signaling pathway of FXR, we assessed the expression of cholesterol 7$\alpha$-hydroxylase (CYP7A1), sterol 12$\alpha$-hydroxylase (CYP8B1), sterol 27-hydroxylase (CYP27A1), and oxysterol 7$\alpha$-hydroxylase (CYP7B1), which are important catalytic enzymes for the synthesis of BAs from cholesterol (36, 37). Then, we found that CYP7A1, a rate-limiting enzyme of the classical pathway of BA synthesis (38), was insignificantly downregulated in LD-fed mice compared with its level in ND-fed mice and that the level of CYP7A1 was markedly reduced with the *L. reuteri* and *L. plantarum* treatments, suggesting that *L. reuteri* and *L. plantarum* treatments can inhibit BA synthesis via the classical pathway (Fig. 3G and H). The mRNA and protein levels of CYP8B1, another important enzyme in the classical pathway, showed no significant change with the LD, *L. reuteri*, and *L. plantarum* interventions (Fig. S2E and F). Furthermore, the mRNA and protein levels of CYP27A1, a rate-limiting enzyme in the alternative pathway, were not decreased in LD-fed mice compared with the levels in ND-fed mice, and the protein level of CYP27A1 was markedly increased with *L. reuteri* and *L. plantarum* treatments, indicating that the alternative pathway was activated in mice with *L. reuteri* and *L. plantarum* treatments (Fig. 3G and H). Finally, the expression of CYP7B1 protein increased in the LD group and reversed with *L. reuteri* treatment (Fig. 3G and H).

Obstruction of the transport of BAs to the gallbladder is an additional risk for the formation of gallstones (39). The multidrug-resistance-associated protein homologs (MRPs) in the basolateral membrane of hepatocytes are responsible for excreting BAs (40). We found that the mRNA expression of MRP4, a transporter that mediates the coefflux of conjugated BAs (41), was significantly decreased by the LD and that the *L. reuteri* and *L. plantarum* treatments markedly upregulated it (Fig. 3I). The mRNA level of hepatic MRP3 was significantly increased in LD-fed mice, and there was a more marked increase with *L. reuteri* or *L. plantarum* treatment (Fig. 3I). Also, the mRNA level of hepatic multidrug resistance protein 2 (MDR2) was not statistically significantly upregulated in LD-fed mice compared with the level in the ND group and there was a significant increase with *L. reuteri* or *L. plantarum* treatment (Fig. 3I). It was suggested that *L. reuteri* and *L. plantarum* intervention improved biliary phospholipid secretion (39). However, the expression of MRP2 was not significantly changed in LD-fed mice regardless of *L. reuteri* or *L. plantarum* treatment (Fig. S2G). Furthermore, the mRNA and protein expression levels of the bile salt export pump (BSEP) were detected and were found to be downregulated in LD-fed mice compared with the levels in ND-fed mice, and this decrease could be reversed by *L. reuteri* and *L. plantarum* treatments (Fig. 3I and J).

To further study the secretion of cholesterol in the liver, we assessed the expression of hepatic cholesterol excretion genes encoding ATP-binding cassette subfamily G members 5 and 8 (ABCG5/8) (42). We first found that the mRNA levels of the ABCG5/8 genes were increased in LD-fed mice compared with the levels in ND-fed mice and that they could be reversed by *L. reuteri* treatment but not *L. plantarum* treatment (Fig. S2H), suggesting that the *L. reuteri* and *L. plantarum* treatments exerted different effects on cholesterol transport. Next, we found that the mRNA and protein levels of sterol-regulatory element binding protein 2 (SREBP2) and SREBP cleavage-activating protein (SCAP) were markedly upregulated in LD-fed mice compared with their levels in ND-fed mice and that the *L. reuteri* and *L. plantarum* treatments significantly reversed these effects (Fig. 2K and L).

Collectively, these data demonstrate that BA synthesis was inhibited through the FXR-FGF15-FGFR4 and FXR-SHP signaling pathways in LD-fed mice and further markedly inhibited by *L. reuteri* and *L. plantarum* treatments. Moreover, the alternative pathway

**FIG 3** Legend (Continued)
quantification of the proteins. (I) Hepatic mRNA levels of *Mrp3*, *Mrp4*, *Mdr2*, and *Bsep*. (J) Protein expression and quantification of hepatic BSEP. (K and L) Hepatic mRNA and protein expression of SCAP and SREBP2 and quantification of the proteins. Data were analyzed by ANOVA along with the *post hoc* Tukey test and are presented as the mean values ± SEM ($n = 6$). *, $0.01 < P \leq 0.05$; **, $0.001 < P \leq 0.01$; ***, $P \leq 0.001$ versus normal diet (ND) group; #, $0.01 < P \leq 0.05$; ##, $0.001 < P \leq 0.01$; ###, $P \leq 0.001$ versus lithogenic diet (LD) group.

was more advantageous than the classical pathway with *L. reuteri* and *L. plantarum* supplements. The excretion of bile acid into the gallbladder was promoted by *L. reuteri* and *L. plantarum*. Consistently, cholesterol synthesis was inhibited by downregulating the expression of SCAP and SREBP2 (43), and cholesterol efflux was suppressed in the *L. reuteri*-treated group.

**L. reuteri and L. plantarum treatments altered the CGS-associated gut microbiota composition in LD-fed mice.** To further validate our hypothesis that *L. reuteri* and *L. plantarum* alleviated CGS by rebuilding the gut microbiota, we collected feces from mice for analysis of the V3-V4 region of the 16S rRNA gene with an Illumina platform. We masked *L. reuteri* and *L. plantarum* from the data set for the distance calculation to avoid the two probiotic strains contributing to the separation and used analysis of similarities (ANOSIM) to compare the similarities of the bacterial communities between samples at the operational taxonomic unit (OTU) level based on Bray-Curtis analysis. Principal coordinate analysis (PCoA), a visual method used to study the differences between bacterial communities, indicated that the OTUs of the LD group were significantly different from those of the ND group and that oral administration of *L. reuteri* and *L. plantarum* shifted the microbial composition of the LD-fed mice toward that of the ND group at the OTU level (Bray-Curtis ANOSIM, $R = 0.6668$, $P = 0.001$) (Fig. 4A). The distance between groups was larger than all the intragroup distances, which indicated that the differences between groups were significantly greater than those within the groups (Fig. 4B). At the phylum level, after 8 weeks of feeding the LD, the average read counts of *Firmicutes*, the most common phylum in most people (44), increased from 5,198 to 14,172. The percentage of *Firmicutes* also increased significantly in the LD group compared with that in the ND group, and the read counts of *Firmicutes* in LD-fed mice decreased from 14,172 to 10,706 and 10,307 with oral *L. reuteri* and *L. plantarum* administration, respectively; however, the percentages in *L. reuteri*- and *L. plantarum*-treated mice did not decrease compared with that in the LD group (Fig. 4C and D). The average read counts and the percentage of *Bacteroidetes* decreased significantly in the LD group compared with those in ND group, and *L. plantarum* treatment decreased them significantly (Fig. 4C and D). Additionally, we found that the read count of *Verrucomicrobia* in the LD group decreased from 2,128 to 249 compared with the count in the ND group, and it increased markedly to 6,720 in *L. plantarum*-treated mice (Fig. 4C). We took a closer look at the *Firmicutes*/*Bacteroidetes* ratio and found that it was significantly increased in LD-fed mice compared with the ratio in ND-fed mice and that the increase was reversed by *L. reuteri* treatment. Contrary to *L. reuteri* treatment, *L. plantarum* treatment did not induce a decrease in the *Firmicutes*/*Bacteroidetes* ratio (Fig. 4D).

At the genus level, the Venn diagrams indicated that the proportion at approximately 48.61% of the total gut bacteria was identical among the four groups (Fig. S3A), revealing that the core microbiota remained stable. We also found that two species, from the *Roseomonas* and *Catabacter* genera, were specific to LD-fed mice (Fig. S3A). At the family level, the percentages of community abundance of *Muribaculaceae* and *Akkermansiaceae* in the community bar plot analysis showed that both were present in lower relative abundances in LD-fed mice than in ND-fed mice (Fig. 4E and F). The abundance of *Muribaculaceae* recovered slightly with *L. reuteri* treatment, and that of *Akkermansiaceae* was greatly increased with *L. plantarum* treatment (Fig. 4E and F). *Muribaculaceae* occupies a major proportion of the community in feces of healthy mice and plays a critical role in improving metabolic disorders (45, 46). In recent years, a number of studies have revealed that *Akkermansiaceae* has an important function in improving metabolic syndrome and reducing intestinal inflammation (47, 48), and our results showed that the reductions in *Akkermansiaceae* in LD-fed mice were reversed by *L. plantarum* treatment. Furthermore, we observed a significantly increased abundance of *Erysipelotrichaceae* in the LD group compared to its abundance in the ND group, which was reversed by *L. reuteri* and *L. plantarum* treatments (Fig. 4E and F). The reduction in *Erysipelotrichaceae* is beneficial to lipid metabolism (49). As the Kruskal-Wallis *H* test bar plot in Fig. 3F shows, *Lactobacillaceae*, which was significantly decreased in LD-fed mice, was increased by *L. reuteri* and *L. plantarum* interventions.

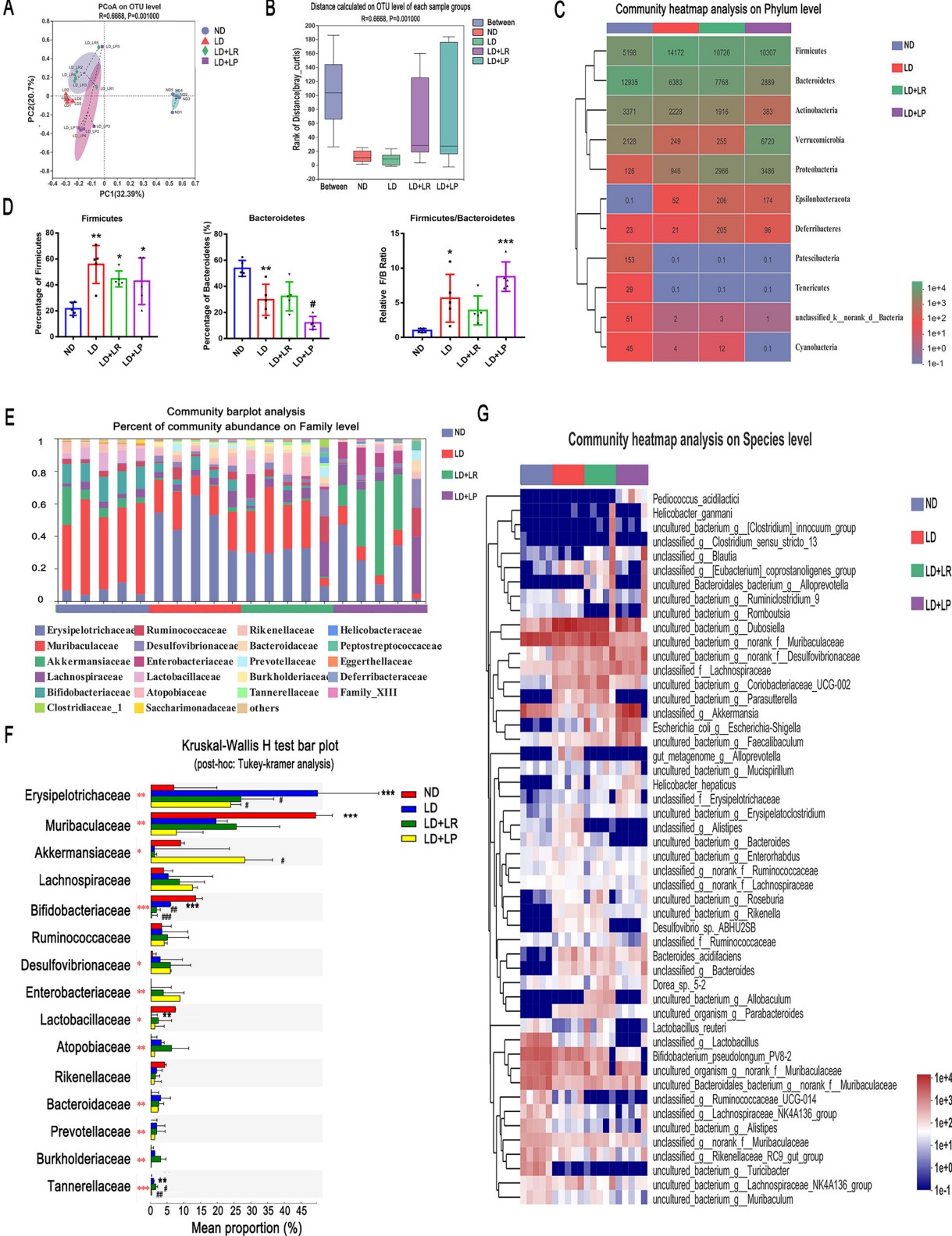

**FIG 4** *L. reuteri* (LR) and *L. plantarum* (LP) treatments changed the CGS-associated gut microbiota composition of LD-fed mice. Fecal samples from ND-fed mice or LD-fed mice with or without *Lactobacillus* treatment were collected at the 8th week for quantification. (A) Principal coordinate analysis (PCoA) of unweighted UniFrac analysis based on the OTU abundances of different groups (Bray-Curtis ANOSIM, $R = 0.6683$, $P = 0.001$). (B) Distance box plots based on the OTU abundances of groups (Bray-Curtis ANOSIM, $R = 0.6668$, $P = 0.001$). (C) Relative abundances of phyla in the gut microbiota of ND-fed mice and LD-fed mice with or without *L. reuteri* or *L. plantarum* treatment. (D) Percentages of *Firmicutes* and *Bacteroidetes* and ratios of *Firmicutes*/*Bacteroidetes* (F/B). (E) The relative abundances of fecal bacteria in each

We further identified the differences in *Lactobacillus* abundances at the species level. The Kruskal-Wallis *H* test bar plot of *Lactobacillus* showed that *L. reuteri* was increased significantly with *L. reuteri* treatment and that *L. plantarum* subsp. plantarum was increased significantly with *L. plantarum* treatment (Fig. S3B). We further examined the species (top 50 abundances) of gut bacteria in mouse feces and found that the abundances of several species were similarly changed in the *L. reuteri* and *L. plantarum* treatments. However, *L. reuteri* and *L. plantarum* modulated some species differently. For example, *Pediococcus acidilactici*, *Akkermansia*, *Erysipelotrichaceae*, and *Erysipelatoclostridium* increased in the *L. plantarum*-treated group but not in the *L. reuteri*-treated group, while *Parasutterella*, *Desulfovibrio* sp. ABHU2SB, and *Allobaculum* decreased in the *L. plantarum*-treated group but not in the *L. reuteri*-treated group (Fig. 4G).

These results indicate that *L. reuteri* and *L. plantarum* treatments manipulated the gut microbiota composition at the phylum, genus, family, and species levels in LD-fed mice, leading to increased numbers of metabolism-promoting flora and inducing a shift in the microbiota abundance patterns from LD-induced metabolic disorders to healthy conditions in different ways.

**Correlation between gut microbiota and host CGS-related parameters.** To further investigate the correlation between the abundances of different gut microbiota and the factors related to the occurrence of CGS, Spearman's correlation analysis was performed between the top 50 most abundant species and families that were changed by the two strains and several CGS-related parameters, including the incidence and grade of CGS (Fig. 5A, Fig. S4A), blood biochemical indexes (Fig. 5B, Fig. S4B), expression levels of FXR pathway-related genes (Fig. 5C, Fig. S4C), and BAs in serum (Fig. 5D, Fig. S4D) and liver (Fig. 5E, Fig. S4E) in all the groups of mice. At the species level, four species and six species were significantly positively correlated with the incidence and grade of CGS, respectively, and four and two species were negatively correlated with them (Fig. 5A). Some opportunistic pathogenic bacteria that were found to be associated with metabolic disorders in previous studies were also found to be positively associated with the incidence or grade of CGS, e.g., *Desulfovibrionaceae* and *Ruminococcaceae* (Fig. 5A) (9, 50). We found that *Akkermansia*, *Muribaculaceae*, and *Muribaculum* were negatively correlated with both the incidence and the grade of CGS (Fig. 5A). At the family level, *Akkermansiaceae* was negatively correlated with both the incidence and the grade of CGS and *Lactobacillaceae* and *Muribaculaceae* were negatively correlated with the incidence of CGS, while *Desulfovibrionaceae* were positively correlated with the incidence of CGS (Fig. S4A). Next, we used Spearman's correlation analysis to investigate the relationships between several serum biochemical indexes and gut microbiota. The results showed that at the species level, 10 of the 50 species were significantly negatively correlated with serum ALT, AST, or ALP, including bacteria belonging to *Akkermansia*, *Muribaculaceae*, *Lactobacillaceae*, and *Muribaculum* (Fig. 5B), and at the family level, 10 of the 50 species were negatively correlated with serum ALT, AST, or ALP, namely, *Muribaculaceae*, *Akkermansiaceae*, *Lactobacillaceae*, etc. (Fig. S4B), suggesting that these bacteria might be involved in repairing liver injury during the process of CGS formation. We found that some gut microbes were significantly positively correlated with serum TG, TC, HDL, LDL, and Glu at the species level, including *Desulfovibrio*, *Dubosiella*, *Roseburia*, and *Parasutterella* (Fig. 5B), and at the family level, *Erysipelotrichaceae*, *Defluviitaleaceae*, *Atopobiaceae*, and others were significantly positively correlated with serum TG, TC, HDL, LDL, and Glu also (Fig. S4B), suggesting that these bacteria might have an impact on metabolic disorders. Furthermore, when we analyzed the correlations between gut microbiota and the mRNA levels of liver *Cyp7a1*,

**FIG 4** Legend (Continued)
group of mice at the family level were analyzed with a community bar plot analysis. (F) Comparison of the top 15 most abundant families in the ND, LD, LD + *L. reuteri*, and LD + *L. plantarum* groups by Kruskal-Wallis *H* test with *post hoc* (Tukey-Kramer) analysis. (G) The top 50 most abundant species in the gut microbiota in different groups are presented in a heatmap. Data are presented as the mean values ± SEM ($n = 5$). \*, $0.01 < P \leq 0.05$; \*\*, $0.001 < P \leq 0.01$; \*\*\*, $P \leq 0.001$ versus normal diet (ND) group; #, $0.01 < P \leq 0.05$; ##, $0.001 < P \leq 0.01$; ###, $P \leq 0.001$ versus lithogenic diet (LD) group. The red asterisks in panel F show statistically significant differences between the groups by Kruskal-Wallis *H* test.

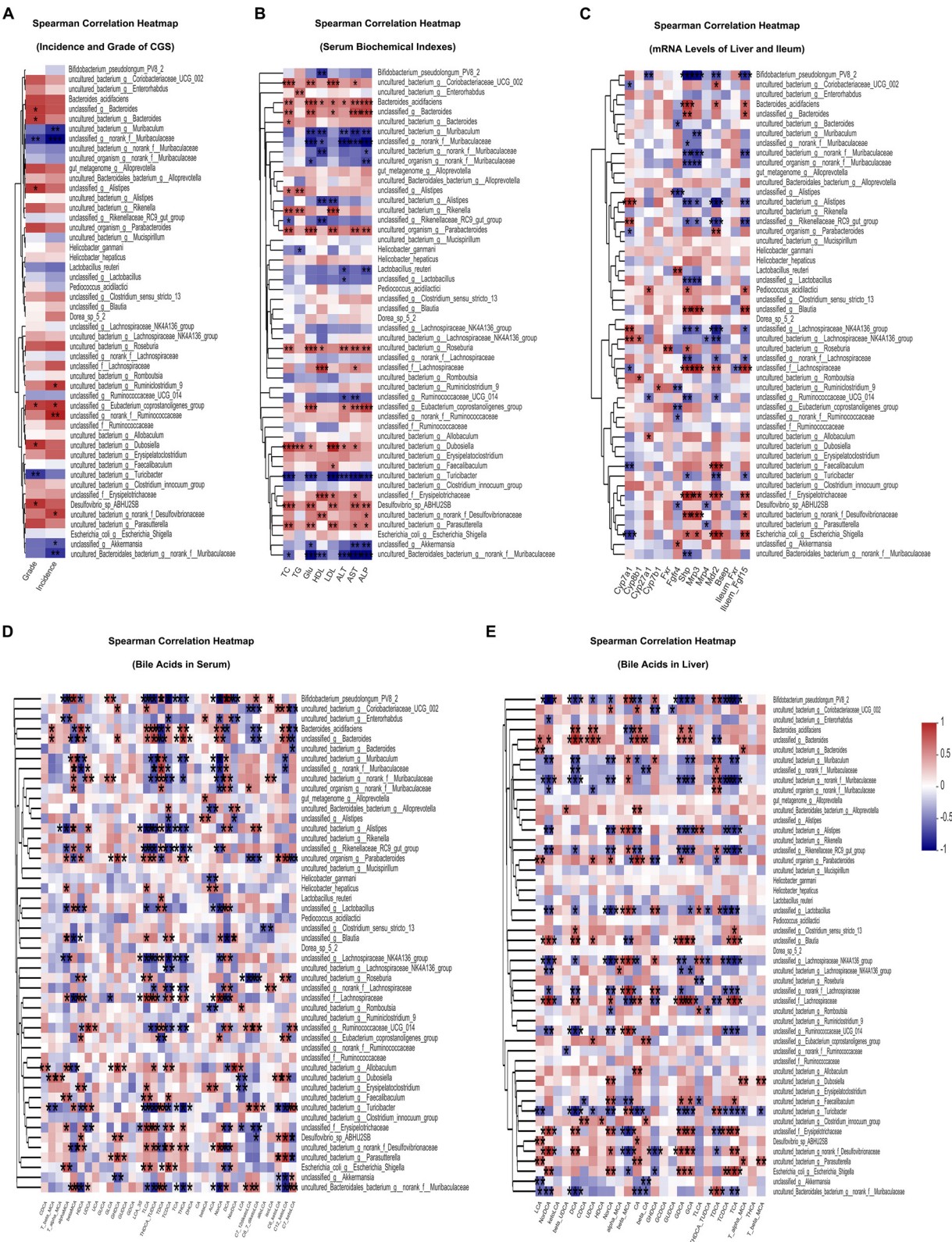

**FIG 5** Correlation between gut microbiota and host CGS-related parameters. (A) Spearman's correlation analysis was performed between the top 50 most abundant species levels and the incidence and grades of CGS. (B) Spearman's correlation analysis was performed between the top 50 most abundant species levels and the serum AST, ALT, ALP, TG, TC, HDL, LDL, and glucose (Glu) levels. (C) Spearman's correlation analysis was performed between the top 50 most abundant species levels and the mRNA levels of liver *Cyp7a1, Cyp8b1, Cyp7b1, Cyp27a1, Mdr2, Mrp3, Mrp4, Bsep, Shp*, hepatic *Fxr*, and *Fgfr4* and ileum *Fxr* and *Fgf15*. (D) Spearman's correlation analysis was performed between the top 50 most abundant species levels and 35 kinds of BAs in the serum of mice. (E) Spearman's correlation analysis was performed between the top 50 most abundant species levels and 26 kinds of BAs in the livers of mice. Red and blue denote positive and negative associations, respectively. *n* = 5. \*, 0.01 < *P* ≤ 0.05; \*\*, 0.001 < *P* ≤ 0.01; \*\*\*, *P* ≤ 0.001; |*R*|≥0.5.

*Cyp7b1*, *Cyp8b1*, *Cyp27a1*, *Fxr*, *Fgfr4*, *Shp*, *Mrp3*, *Mrp4*, *Mdr2*, and *Bsep* and ileum *Fxr* and *Fgf15*, we found that *Akkermansia* and *Lactobacillus* were significantly positively correlated with liver *Fgfr4* at the species level (Fig. 5C). The abundance of *Enterobacteriaceae* was significantly positively correlated with the ileum FXR signaling-related gene ileum *Fgf15* and genes related to BA export into bile, including *Bsep*, *Mdr2*, and *Mrp3*, at the species and family levels (Fig. 5C, Fig. S4C). However, *Muribaculaceae* showed a negative correlation with liver *Shp* (Fig. 5C, Fig. S4C). These results suggested that the gut microbiota have an impact on the expression profile of genes involved in FXR signaling pathways and BA synthesis and exportation. Finally, we found that several gut bacteria were significantly correlated with the composition of BAs in serum and liver, suggesting that gut microbiota played a vital role in BA metabolism (Fig. 5D and E, Fig. S4D and E). These data demonstrated that the CGS phenotype, serum biomarkers, FXR signaling pathway-related genes, and the composition of BAs were significantly correlated with gut microbiota and that the protective effects on CGS formation exerted by *L. reuteri* and *L. plantarum* treatments might be due to their regulating the gut microbiota.

**Inhibition of FXR activation attenuated the protective effects of *L. reuteri* and *L. plantarum* in LD-fed mice.** The data presented above indicate a critical involvement of FXR in *L. reuteri*- and *L. plantarum*-mediated prevention of gallstones in LD-fed mice. We hypothesized that the beneficial effects of *L. reuteri* and *L. plantarum* treatments could be abolished by the inhibition of FXR activation. To test this hypothesis, mice fed the ND or LD and administered the *L. reuteri* or *L. plantarum* treatments for 8 weeks were orally administered Z/E-guggulsterone (Z-Gu), a global FXR inhibitor, or glycine-$\beta$-muricholic acid (Gly-MCA), an intestine-specific FXR inhibitor (17). Compared to the round gallstones and sticky bile observed in the gallbladders of LD-fed mice, we found only a few leaflet crystals or some cholesterol particles in the gallbladders of *L. reuteri*- and *L. plantarum*-treated mice, and the incidence and the grade of gallstones could be reduced by *L. reuteri* and *L. plantarum* treatments. However, Z-Gu almost completely reversed these protective effects, and granular stones filled the gallbladder. Gly-MCA also weakened the protective effects in reducing gallstone formation, and several leaflet crystals were suspended in bile. Under the Z-Gu intervention, the quantity and quality of gallstones were higher, and the incidence of gallstones was higher than in the Gly-MCA intervention (Fig. 6A and B), suggesting that it was the global FXR that reduced CGS formation, not the intestinal FXR alone. Then, we calculated the ratio of liver weight to body weight in mice and found that inhibiting FXR by Z-Gu and Gly-MCA treatment could significantly reverse the *L. reuteri*-induced reduction of the liver weight-to-body weight ratio, but there were no significant changes in *L. plantarum*-treated mice (Fig. 6C). Hematoxylin and eosin (H&E) staining and oil red O (ORO) staining were performed to examine hepatic steatosis. Consistently, we found that Z-Gu abolished the protective effects of the *L. reuteri* and *L. plantarum* treatments and that Gly-MCA slightly blunted these effects (Fig. S5A). The changes in relative lipid content shown by liver oil red O staining were consistent with the tissue slice staining results (Fig. S5B), suggesting that global FXR was necessary for *L. reuteri* or *L. plantarum* to improve liver fat metabolism and that intestine-specific FXR played only a synergistic role. In addition, the *L. reuteri*- and *L. plantarum*-induced reductions in the volume of the gallbladder and the thickness of the gallbladder's tunica adventitia were markedly reversed by the Z-Gu and Gly-MCA interventions, and the results showed that inhibiting global FXR, but not intestine-specific FXR, could make the gallbladder wall thicker than LD feeding alone (Fig. 6D and E). A likely reason is that inhibiting global FXR instead of intestine-specific FXR may lead to gallbladder wall thickening.

Serum ALT, AST, and ALP were analyzed, and we found that both the Z-Gu and Gly-MCA interventions reversed the decreases in ALT, AST, and ALP caused by *L. reuteri* or *L. plantarum* treatment and that Z-Gu was more effective (Fig. 6F), indicating that inhibiting global FXR may almost completely abolish the protective effects against liver injury from *L. reuteri* and *L. plantarum* treatments in LD-fed mice. Moreover, we found that the *L. reuteri* and *L. plantarum* treatments effectively reduced serum TC and that the reduction was significantly reversed by the intestine-specific FXR inhibitor Gly-MCA

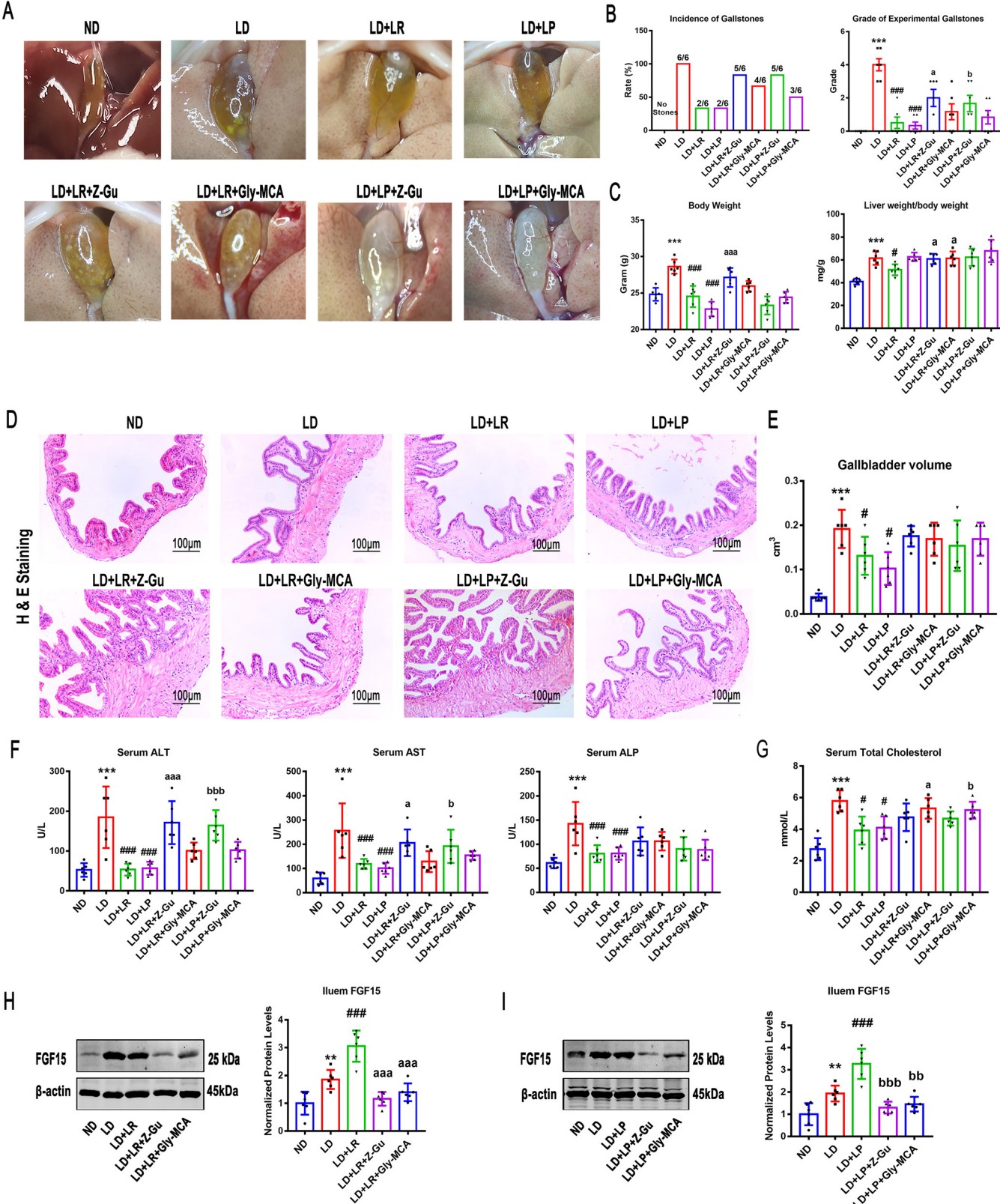

**FIG 6** Inhibition of FXR activation attenuated the protective effects of *L. reuteri* (LR) and *L. plantarum* (LP) in LD-fed mice. Forty-eight mice were randomly assigned to 8 groups fed the ND or LD with or without *L. reuteri* or *L. plantarum* treatment (10⁹ CFU/day). At the beginning of the *L. reuteri* or *L. plantarum* treatment, the mice were gavaged with (Z/E)-guggulsterone (Z-Gu), a global FXR inhibitor, or glycine-*β*-muricholic acid (Gly-MCA), an intestine-specific FXR inhibitor. (A) Gross appearance of gallbladders and gallstones of mice administered different treatments. (B) Percentage of gallstone incidence in each

(Fig. 6G). However, the results showed that the reductions of serum TC and TG by *L. reuteri* and *L. plantarum* were not significantly affected by the global FXR inhibitor (Fig. 6G, Fig. S5C). Consistent with previous research results (51), the Z-Gu and Gly-MCA interventions abolished the protective effect of *L. reuteri* and *L. plantarum* treatments on serum glucose (Fig. S5D). Furthermore, the protein expression levels of FGF15 were detected and were markedly decreased with Z-Gu and Gly-MCA treatments compared with the levels in LD-fed mice with or without *L. reuteri* and *L. plantarum* treatments.

Taken together, these data demonstrate that the *L. reuteri* and *L. plantarum* treatments ameliorate experimental gallstones at least partly in an FXR-dependent manner.

## DISCUSSION

Although CGS patients and mice have low abundances of *Lactobacillus* (10, 52), whether *Lactobacillus* strains can mitigate CGS and the mechanism by which they might do so remain unclear. In this study, we found (i) that one *L. reuteri* strain and one *L. plantarum* strain significantly alleviated CGS and its related fat deposition in the liver and serum and insulin resistance, with critical roles in their effects being played by both global and intestine-specific FXR, especially the global FXR, (ii) that the two *Lactobacillus* strains altered the serum and liver BA pools, especially by downregulating FXR antagonists T-$\alpha$-MCA and T-$\beta$-MCA, and decreased BA synthesis and enhanced BA transport in the liver by activating the ileum and liver FXR signaling pathways, and (iii) that the two strains significantly improved LD-induced CGS-associated gut microbiota dysbiosis, including the increases in *Muribaculaceae* by *L. reuteri* and in *Akkermansia* by *L. plantarum*. All these results suggested that *Lactobacillus* might prevent CGS through the FXR signaling pathways.

Our study showed that one of the most important mechanisms by which the *L. reuteri* and *L. plantarum* treatments reduced the incidence of gallstones was the activation of the ileal FXR-FGF15-FGFR4 and hepatic FXR-SHP signaling pathways. We found that the levels of T-$\alpha$-MCA and T-$\beta$-MCA were markedly increased with LD-fed and that the FXR signaling pathways were inhibited. Cai and colleagues showed that the mRNA level of FXR in patients with cholelithiasis was lower than that in control individuals (21), suggesting that the inhibition of FXR signaling is very likely to promote the onset of CGS. Furthermore, the expression of FGFR4 was significantly decreased, suggesting that the ileal FXR-FGF15-FGFR4 signaling pathway was partially blocked. These data show that the intestinal FXR activated by the high concentration of cholic acid (CA) in the LD was not sufficient to activate hepatic FGFR4 signals via FGF15. Excessive intake of CA resulted in severe disturbance of BA pools in mice. Lack of FXR signaling is one of the major risk factors for CGS formation because it weakens the FXR-driven negative feedback regulation of BA synthesis and transportation and leads to a lithogenic BA pool (20). Our results showed that the *L. reuteri* and *L. plantarum* treatments decreased the levels of T-$\alpha$-MCA and T-$\beta$-MCA and increased the levels of TCA and TCDCA to activate FXR signaling and improve several metabolic effects that could reduce the risk of CGS formation. The upregulation of the expression of ileal FGF15 and hepatic SHP in *L. reuteri*- and *L. plantarum*-treated mice was caused by the activation of FXR, which downregulated the level of CYP7A1 to inhibit the classical pathway for BA synthesis and inhibit the expression of SREBP2 and SCAP to reduce cholesterol synthesis. Moreover, it has been reported that increased SHP and FGF15 can inhibit intestinal cholesterol absorption (53). Furthermore, the hydrophilicity of BAs was upregulated by

**FIG 6** Legend (Continued)

group of mice and grade of experimental CGS, based on visualized cholelithiasis. (C) Body weights of mice on the last day and ratios of liver weight to body weight. (D) Representative images of H&E-stained gallbladder sections ($\times$200). (E) Gallbladder volumes estimated by the length, diameter, and circumference of the gallbladders. (F and G) Serum ALT, AST, ALP, and TC were determined by using a Siemens fast automatic biochemical analyzer (Advia 2400). (H and I) Protein expression and quantification of ileal FGF15 with or without *L. reuteri* (H) or *L. plantarum* (I) treatment. Data were analyzed by ANOVA along with the *post hoc* Tukey test and are presented as the mean values $\pm$ SEM ($n = 6$). *, $0.01 < P \leq 0.05$; **, $0.001 < P \leq 0.01$; ***, $P \leq 0.001$ versus normal diet (ND) group; #, $0.01 < P \leq 0.05$; ##, $0.001 < P \leq 0.01$; ###, $P \leq 0.001$ versus lithogenic diet (LD) group; a, $0.01 < P \leq 0.05$; aa, $0.001 < P \leq 0.01$; aaa, $P \leq 0.001$ versus LD + *L. reuteri*; b, $0.01 < P \leq 0.05$; bb, $0.001 < P \leq 0.01$; bbb, $P \leq 0.001$ versus LD + *L. plantarum*.

increasing the hepatic taurine-conjugated BAs in the *L. reuteri*- and *L. plantarum*-treated groups. In addition, we observed that the levels of hepatic MRP3, MRP4, MDR2, and BSEP were significantly upregulated by the *L. reuteri* and *L. plantarum* treatments, indicating that BA drainage to the gallbladder increased to prevent the precipitation of cholesterol from bile. These results suggest that the *L. reuteri* and *L. plantarum* treatments effectively activated FXR signaling to regulate BA and cholesterol metabolism, which could improve the balance between BA and cholesterol.

Our data showed that the protective effects of the two strains on CGS were almost completely blocked by the global FXR inhibitor and the intestine-specific FXR inhibitor alone. Moreover, global FXR performed better in reducing hepatic steatosis and gallbladder wall thickness. This phenomenon suggested that hepatic and ileal FXR signaling both played major roles in protecting the host from CGS after the administration of the two *Lactobacillus* strains. The FXR-FGF15-FGFR4 and FXR-SHP signaling pathways were activated by *Lactobacillus* via downregulating T-$\alpha$-MCA and T-$\beta$-MCA and upregulating TCDCA in the serum and liver. Zhang and colleagues showed that *Lactobacillus casei* strain YRL577 alleviated nonalcoholic fatty liver disease in mice by increasing the intestinal FXR-FGF15 pathway (54), and Liu and colleagues found that FXR in the liver and intestine activated by *Lactobacillus rhamnosus* GG had a synergistic effect to reduce cholestasis (17). These conclusions support our results showing that liver FXR signaling, as well as intestinal FXR signaling, might play a pivotal role in CGS prevention. In addition, Kim and colleagues showed differential regulation of BA homeostasis by FXR in the liver and intestine and used tissue-specific *Fxr*-null mice to suggest the more effective regulation of BA metabolism-related enzymes by liver FXR (55). This conclusion also supports our results showing that only the liver FXR signaling pathway activated by *L. reuteri* and *L. plantarum* was capable of preventing the hepatic steatosis and gallbladder wall hyperplasia that are part of the process of CGS formation.

Previous studies have proven that the gut microbiota and BA metabolism are mutually regulated by each other (31, 56). In our study, we found that the gut microbiota was disrupted and that several metabolic disorders were present in LD-fed mice. *L. reuteri* and *L. plantarum* improved these disorders, sometimes in different ways. *L. reuteri* treatment mainly played the role of increasing the diversity of the gut microbiota and restoring the proportion of *Muribaculaceae*. It is worth noting that *L. plantarum* treatment did not affect the diversity of the gut microbiota but greatly increased the proportion of *Akkermansia* to improve metabolic disorders. This phenomenon is consistent with our previous research results (57). Moreover, we found that *Akkermansia* was significantly positively correlated with FXR signaling pathway-related genes (*Shp* and *Fgf15*), the gene encoding the rate-limiting enzyme in the alternative pathway of BA metabolism (*Cyp27a1*), and a gene involved in BA exportation (*Bsep*), indicating that *Akkermansia* in *L. plantarum*-treated mice is involved in improving BA metabolism and transportation. The results showed that the increase of *Akkermansia* was several times that of *L. plantarum* in *L. plantarum*-treat mice. A reason for this phenomenon may be that *L. plantarum* produces some metabolites that promote the growth of *Akkermansia*. Dao and colleagues showed that *Akkermansia* abundance was associated with a healthier metabolic status (48). Depommier and colleagues demonstrated that supplementation with *Akkermansia* could improve obesity-related disorders and liver injury while maintaining the gut microbiome structure balance (58). This phenomenon has revealed that some probiotics might show their effects by modulating the inherent host gut microbiota. Even though *L. reuteri* and *L. plantarum* are *Lactobacillus* strains and both have protective effects against CGS formation, the influences of the two *Lactobacillus* strains on the gut microbiota were quite different, suggesting functional redundancy of the gut microbiota.

Hu and colleagues identified enrichment of *Desulfovibrionales* in feces from patients with cholesterol GS, as well as in gallstone-susceptible mice (9), which is consistent with what we found in gut microbiota of LD-fed mice. Furthermore, we found that the *Desulfovibrionaceae* level was significantly positively correlated with

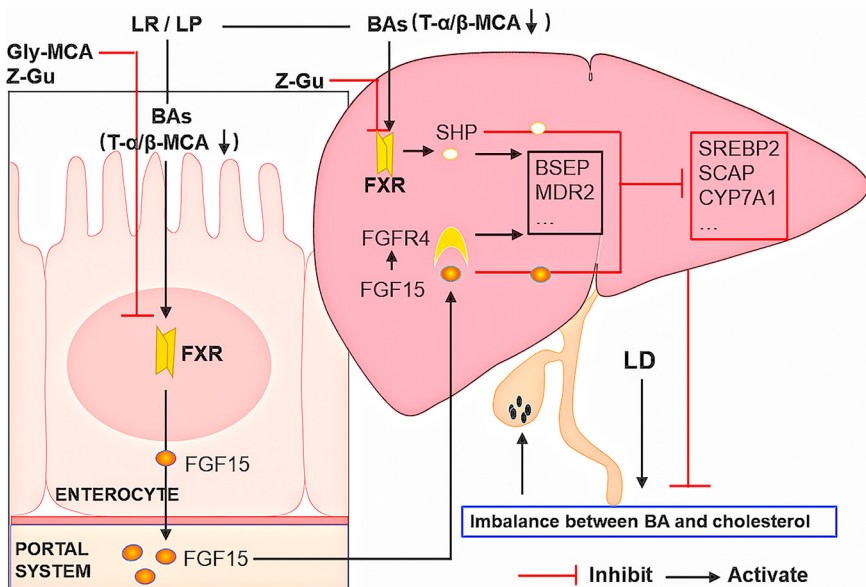

**FIG 7** Schematic diagram summarizing the mechanisms by which *L. reuteri* and *L. plantarum* prevent CGS formation.

the incidence and the grade of CGS and was positively correlated with serum AST, ALP, and glucose. Neither *L. reuteri* nor *L. plantarum* treatment could reduce the abundance of *Desulfovibrionaceae* at the family level. Therefore, the level of *Desulfovibrionaceae* might not be relevant to the reduction of the formation of CGS by *L. reuteri* and *L. plantarum* treatments. In addition, our study found that *Roseomonas* and *Catabacter* were specific to LD-fed mice and that the two genera were always implicated in infection (59, 60). To our knowledge, this is the first study to find a potential relationship between these two genera and CGS.

In summary, we demonstrated that gut microbiota-targeted *L. reuteri* and *L. plantarum* treatments reduced the concentrations of T-$\alpha$-MCA and T-$\beta$-MCA and promoted FXR activation to reduce CGS formation. The reduction of T-$\alpha$-MCA and T-$\beta$-MCA relieved the competitive inhibition of FXR and increased the levels of ileal FGF15 and hepatic SHP to inhibit BA and cholesterol synthesis and promote the excretion of BAs to the gallbladder. Furthermore, the increase of hepatic FGFR4 promoted the ileal FXR-FGF15 signal transduction to inhibit the classical pathway of BA synthesis (Fig. 7). Although additional research is needed before translation to patients, supplementation with *L. reuteri* or *L. plantarum* could be a promising therapeutic strategy for treating CGS or preventing the recurrence of CGS.

## MATERIALS AND METHODS

**Mouse strains and treatments.** Male C57BL/6J mice (about 8 weeks old and not siblings) weighing 20 to 22 g were purchased from Shanghai SLAC Laboratory Animal Co. Ltd. (Shanghai, China). All mice were maintained under 12-h light/dark cycles at 22°C with unlimited access to a standard rodent diet and water and were housed under specific-pathogen-free (SPF) conditions. The mice were allowed to acclimate for at least 1 week before experiments. All animal experiments were approved (SYXK 2013-0050) by the Animal Ethics Committee of Shanghai Jiao Tong University School of Medicine, Shanghai, China, and were conducted under the "3R" principle (reduction, replacement, and refinement). A lithogenic-diet-induced mouse model of CGS was used in this study. Mice were fed a standard chow diet (normal diet [ND], containing 0.02% cholesterol) or a lithogenic diet (LD, containing 15% butter fat, 28% casin, 1.5% cholesterol, and 0.5% cholic acid) for 8 weeks (61). To determine the effects of the administration of *Limosilactobacillus reuteri* strain CGMCC 17942 (*L. reuteri* herein) and *Lactiplantibacillus plantarum* strain CGMCC 14407 (*L. plantarum* herein), 48 mice were randomly allocated into four groups and 12 mice in each group were randomly divided into four plastic cages (*n* = 12 per group and *n* = 3 per cage). The mice in the ND group, fed the normal diet, received 200 $\mu$L of normal saline, which served as the control, and the mice in the LD group, fed the lithogenic diet, received 200 $\mu$L of normal saline for 8 weeks. The mice in the *L. reuteri* group and the *L. plantarum* group, which were fed the LD, received 200 $\mu$L of a bacterial suspension of either *L. plantarum* or *L. reuteri* at a dosage of 10$^9$ CFU/day for

8 weeks. Five mice were randomly selected from each group to have serum and fecal samples collected for bile acid analysis and gut microbiota sequencing. Next, to further verify the role of FXR, 48 mice were randomly allocated into 8 groups. Six mice in each group were randomly divided into two plastic cages ($n = 6$ per group and $n = 3$ per cage), and the mice in four of the groups were treated as described above. The mice in another two groups were fed the LD and orally administered 200 $\mu$L of *L. reuteri* or *L. plantarum* suspension along with the intestine-specific FXR inhibitor Gly-$\beta$-MCA (10 mg/kg of body weight), and the mice in the final two groups were fed the LD and orally administered 200 $\mu$L of *L. reuteri* or *L. plantarum* suspension along with the global FXR antagonist Z/E-guggulsterone (10 mg/kg) for 8 weeks (17).

**Probiotic preparations.** *L. plantarum* and *L. reuteri* were incubated with de Man-Rogosa-Sharpe liquid medium (MRS broth; BD, USA) under anaerobic conditions at 37°C for 24 h to reach the early stationary phase. The cultures were centrifuged at 5,000 $\times$ *g* for 5 min at 4°C, and bacterial cells were resuspended in sterilized saline solution ($10^9$ cells/200 $\mu$L) and stored at $-80$°C as a stock solution. Each aliquot was thawed for 1 h to room temperature before it was administered to a mouse by oral gavage. To determine the number of bacterial cells, serial dilutions of the bacterial suspensions were inoculated on fresh MRS agar plates under anaerobic conditions.

**Sample collection.** The mice were treated for 8 weeks after a week of adaptation. The body weight and food intake of the mice were checked once a week (57). At the beginning of the 8th week, fecal samples were collected from the mice and stored in liquid nitrogen for RNA extraction and subsequent quantification of the gut microbiota as previously described (62). At the end of the 8th week, the mice were fasted for 5 h but allowed free access to water. Blood samples of mice were collected by heart puncture and centrifuged at 400 $\times$ *g* for 20 min at 4°C for subsequent biochemical analysis. The animals were sacrificed by exsanguination after being anaesthetized with pentobarbital. The liver, gallbladder, and ileum were harvested and divided into two parts; one part was immediately snap-frozen in liquid nitrogen and the other part was fixed in 4% neutral paraformaldehyde at room temperature for 24 h.

**Histology.** The liver and gallbladder specimens fixed in paraformaldehyde were used for paraffin embedding. Paraffin-embedded sections (4 $\mu$m) were stained with hematoxylin and eosin (H&E), and a 20$\times$ objective was used over 5 separate fields by two expert liver pathologists who were blinded to the treatment groups to identify the percentages of hepatocytes affected by steatosis. Hepatic lipid accumulation was measured by oil red O (ORO) staining as described previously (63). Frozen liver sections (6 $\mu$m) were stained with ORO lipid stain (Abcam, USA), and the percentages of hepatocytes involved in steatosis were assessed by relative lipid content, which was quantified by ImageJ.

**Bile acid analysis.** Bile acid (BA) concentrations were determined using negative electrospray liquid chromatography-tandem mass spectrometry (LC-MS/MS). Chromatographic separation was achieved on an Acquity ultraperformance liquid chromatography (UPLC) BEH $C_{18}$ column (100-mm inner diameter, 1.7 $\mu$m; Waters Corp.). Serum and liver tissues were accurately measured, ethanol ($-20$°C) was added for precipitation, and the mixture vortex mixed for 60 s and then centrifuged. For liver tissues, 100-mg glass beads were added and oscillated before centrifuging. The supernatant was filtered through a 0.22-$\mu$m membrane before LC-MS/MS. The mobile phase consisted of a mixture of 0.1% formic acid in water (A) and 0.1% formic acid in acetonitrile (B). Gradient elution was applied for the following phases: 0 to 4 min, 25% B; 4 to 9 min, 25 to 30% B; 9 to 14 min, 30 to 36% B; 14 to 18 min, 36 to 38% B; 18 to 24 min, 38 to 50% B; 24 to 32 min, 50 to 75% B; 32 to 35 min, 75 to 100% B; and 35 to 38 min, 100 to 25% B. The flow rate was 0.25 mL/min.

The 40 kinds of BA standards (Yuanye Bio-Technology, Shanghai, China) were prepared in methanol and equally mixed to generate working standard solutions (clear without any precipitation). The working standard solutions were determined by LC-MS/MS, and the total ion chromatography (TIC) analysis of the working standard solutions is shown in Fig. S6. We used a signal-to-noise ratio of 10 ($S/N = 10$) to determine the limit of quantitation (LOQ) of BA standards and selected the appropriate standard curve concentration to make the concentration curve, as shown in Table S2. The supplier numbers, CAS numbers, and International Chemical Identifier (InChI) keys for the standard analytes, their retention times, and the mass transition ion pairs (*m/z*) (precursor ion and product ion) are presented in Table S3. Working standard solutions were freshly prepared before using LC-MS/MS. The concentration of the working standard solution was taken as the abscissa and the peak area as the ordinate to examine the linear range and draw the standard curve. The linear regression equation was obtained for each BA (correlation coefficient *r* of >0.99) and used for performing quantitative analysis on all samples.

**Bile analysis and CSI calculation.** The levels of metabolites in bile were measured by enzymatic assays using a microplate reader (BioTek, Winooski, VT, USA). The levels of total cholesterol (TC) and bile acids (BAs) in bile were measured following the instructions in the kit (Jiancheng Bioengineering Institute, Nanjing, China). The phospholipid levels in bile were measured following the instructions in the kit (Wako Pure Chemicals, Osaka, Japan). The cholesterol saturation index calculation (CSI) for gallbladder bile was performed as described previously (64).

**Immunofluorescence.** Immunofluorescence was used to detect ileal FGF15 in the paraffin-embedded ileum tissue sections. The process of dewaxing, hydration, antigen retrieval, permeabilization, blocking of nonspecific binding, and incubation with antibodies was performed as previously described (62). The sections were imaged with a Leica TCS SP8 confocal microscope ($\times$400).

**Western blotting.** Total proteins from liver and ileum tissues were extracted as previously described (62). Total proteins (40 $\mu$g per lane) were separated by 10% SDS-PAGE and then electrotransferred to nitrocellulose membranes. The membranes were incubated with primary antibodies against FXR (NR1H4, 1:1,000; Abcam, USA), SREBP2 (1:1,000; Abcam, USA), CYP7A1 (1:1,000; Abcam, USA), CYP8B1 (1:1,000; Abcam, USA), CYP7B1 (1:1,000; Proteintech Group, USA), CYP27A1 (1:1,000; Proteintech Group,

**TABLE 1** Primers used for PCR

| Mouse gene | Primer direction | Sequence |
|---|---|---|
| *Fgf15* | Forward | 5′-CGGTCGCTCTGAAGACGATTGC-3′ |
| | Reverse | 5′-TACATCCTCCACCATCCTGAACGG-3′ |
| *Fgfr4* | Forward | 5′-GCCCCTGTACGTGATTGTGG-3′ |
| | Reverse | 5′-ATCCATTTGACTGGCAGGCG-3′ |
| *Shp* | Forward | 5′-TCCTAGCCAAGACAGTAGCCTTCC-3′ |
| | Reverse | 5′-TACCGCTGCTGGCTTCCTCTAG-3′ |
| *Mrp3* | Forward | 5′-TGAGATCGTCATTGATGGGC-3′ |
| | Reverse | 5′-AGCTGAGAGCGCAGGTCG-3′ |
| *Mrp4* | Forward | 5′-TTAGATGGGCCTCTGGTTCT-3′ |
| | Reverse | 5′-GCCCACAATTCCAACCTTT-3′ |
| *Mdr2* | Forward | 5′-GGATGGTGACTGTGGGCTGAT-3′ |
| | Reverse | 5′-GGCTGTTCTCCCTTCTCATGG-3′ |
| *Bsep* | Forward | 5′-GCTGCCAAGGATGCTAATGC-3′ |
| | Reverse | 5′-CTACCCTTTGCTTCTGCCCA-3′ |
| *Cyp7a1* | Forward | 5′-GCTAAGACGCACCTCGTGATCC-3′ |
| | Reverse | 5′-CCGCAGAGCCTCCTTGATGATG-3′ |
| *Cyp27a1* | Forward | 5′-ATTAAGGAGACCCTGCGCCT-3′ |
| | Reverse | 5′-AGGCAAGACCGAACCCCATA-3′ |
| *Cyp7b1* | Forward | 5′-TGGCTTCCTTATCTTGGCATGGC-3′ |
| | Reverse | 5′-TCGCTGATAATCGGCTGCTGAAC-3′ |
| *Scap* | Forward | 5′-GGCAATCTCATCGTGG-3′ |
| | Reverse | 5′-ATGGTCTTGGCTCCCT-3′ |
| *Srebp2* | Forward | 5′-AGCAGTGGCAGAGGCAACAATG-3′ |
| | Reverse | 5′-CTGGTCGCTGCGTTCTGGTATATC-3′ |

USA), BSEP (1:200; Santa Cruz Biotechnology, USA), and $\beta$-actin (1:1,000; Cell Signaling Technology, USA) overnight at 4°C. After washing with 0.1% Tween 20–phosphate buffer solution (PBST) three times, the membranes were incubated with goat anti-mouse or goat anti-rabbit IRDye 700 or 800 calcofluor white (CW)-labeled secondary antibodies for 1 h at 37°C and imaged with an Odyssey infrared scanner (LI-COR, Lincoln, NE, USA). Quantification was performed using the LI-COR software Image Studio.

**RNA extraction and quantitative real-time PCR.** Total RNA from liver and ileum tissues was extracted using TRIzol (Invitrogen, CA, USA) homogenization followed by isopropanol incubation as previously described, and the purity of RNA products was determined to be between 1.8 and 2.0 according to the 260/280 ratio. One thousand nanograms of RNA was subjected to reverse transcription (RT) to generate cDNA using a commercial PrimeScript RT reagent kit (TaKaRa, Japan). The synthesized cDNA was used for quantitative PCR (qPCR) to determine the relative expression of targeted genes using gene-specific, intron-spanning primers (Table 1). Hieff qPCR SYBR green master mix (Yeasen Biotech, Shanghai, China) was used to perform qPCR to analyze mRNA transcripts, and all reactions were performed using the ABI Prism 7900HT sequence detection system (Applied Biosystems, CA, USA). The fold changes in the expression of each target gene were compared to the amount of the housekeeping gene $\beta$-actin using the cycle threshold ($2^{-\Delta\Delta CT}$) method and are represented as the fold change relative to the value for the control group. Each target gene analyzed in the tissues was analyzed in triplicate in experiments that were repeated 3 times.

**DNA extraction and bacterial sequencing.** At the end of the experiment, we randomly selected 5 mice from the 12 mice in each group to obtain stool samples for gut microbiota detection; the stool samples from the other mice were frozen in liquid nitrogen. Mouse stool samples were collected within 15 min of defecation and stored in liquid nitrogen within 1 h. Bacteria DNA was extracted using QIAamp DNA stool kits (Qiagen, CA, USA). The 16S rRNA gene was PCR amplified using primers binding to the V3-V4 regions. The resulting amplicons were extracted from 2% agarose gels, purified using gel extraction (AxyPrep DNA gel extraction kit; Axygen Biosciences, Union City, CA, USA), and then quantified and sequenced on the Illumina MiSeq platform (Illumina, San Diego, USA) with paired-end 300-nucleotide reads. The 16S rRNA sequencing data were analyzed by using the Quantitative Insights Into Microbial Ecology (QIME) platform (version 1.9.1).

**Bioinformatics.** The raw 16S rRNA gene sequencing reads were demultiplexed, quality filtered by using fastp version 0.20.0, and merged by using FLASH version 1.2.7 (65, 66) with the following criteria: (i) no contaminant sequences were allowed; (ii) 300-bp reads were truncated at any site receiving an average quality score of <20 over a 50-bp sliding window, with parts of truncated reads shorter than 50 bp and parts of reads containing ambiguous characters both being discarded; (iii) only overlapping sequences longer than 10 bp were assembled, according to their overlapped sequence, with a maximum mismatch ratio of the overlap region of 0.2 and reads that could not be assembled being discarded; and (iv) samples were distinguished according to the barcode and primers, the sequence direction adjusted, exact barcode matching performed, and 2-nucleotide mismatch in primer matching allowed. Operational taxonomic units (OTUs) with a 97% similarity cutoff (67) were clustered using UPARSE version 7.1, and chimeric sequences were identified and removed.

**Statistical analysis.** The nonmicrobiome data obtained were analyzed statistically to determine the level of significance using one-way analysis of variance (ANOVA) along with the *post hoc* Tukey test. Tukey's *post hoc* test was performed for data with $F$ at a $P$ value of <0.05 and no significant variance inhomogeneity. Statistical analysis was performed using SPSS 24 (SPSS, Inc., Chicago, IL, USA). GraphPad Prism 7.0 (GraphPad Software, Inc., La Jolla, CA, USA) was used to shape the experimental data, and the data are presented as the mean values $\pm$ standard errors of the means (SEM) ($n = 5$ to 12). A $P$ value of <0.05 was considered statistically significant. For the gut microbiota data analysis, the abundances of various taxons were evenly subsampled before analysis and the taxonomy of each representative OTU sequence was analyzed by RDP Classifier version 2.2 against the 16S rRNA database using a confidence threshold of 0.7 (68). Principal coordinate analysis (PCoA) based on Bray-Curtis dissimilarity was performed to provide an overview of gut microbial dynamics in response to LD and *L. reuteri* or *L. plantarum* treatments. Similarities of gut microbiota between samples were compared by ANOSIM and Adonis based on Bray-Curtis at the phylum level. Statistical comparisons of the relative abundances in different groups were analyzed by using the Kruskal-Wallis $H$ test and Tukey-Kramer test (with 95% confidence intervals and $P$ values of <0.05; multiple-testing $P$ values were adjusted with the false discovery rate [FDR]). The relationships between the top 50 most abundant organisms at the species level, the incidence and grades of CGS, and the serum BAs, liver BAs, serum biochemical indexes, and levels of CGS-related mRNA expression were analyzed by Spearman's correlation analysis, and the read counts of the abundances of various taxons were transformed to the centered log ratio for Spearman correlation analysis (69). The Spearman's correlation coefficient was $|R| \geq 0.5$. $P$ values are as follows: $0.01 < P \leq 0.05$, $0.001 < P \leq 0.01$, and $P \leq 0.001$.

**Reagents.** Z/E-guggulsterone and glycine-$\beta$-muricholic acid were purchased from MedChemExpress (MCE, USA). The primary antibodies against CYP7A1, CYP8B1, and FXR were purchased from Abcam, USA. $\beta$-Actin was purchased from Cell Signaling Technology (CST, USA). The primary antibodies against CYP7B1, CYP27A1, and FGFR4 were from Proteintech Group, USA. The primary antibodies against FGF15 and BSEP were from Santa Cruz Biotechnology, USA, and all the other reagents were from Sigma-Aldrich Chemical (St. Louis, MO, USA).

**Data availability.** The sequence files and metadata for all samples used in this study have been deposited into the NCBI Sequence Read Archive (SRA) database (accession number PRJNA680858).

## SUPPLEMENTAL MATERIAL

Supplemental material is available online only.
**SUPPLEMENTAL FILE 1**, PDF file, 1.3 MB.

## ACKNOWLEDGMENTS

We thank American Journal Experts for the expert linguistic services provided.

We declare that there is no conflict of interest.

This work was sponsored by grants from the National Natural Science Foundation of China to X. Wan (grant number 81870452) and J. Wang (grant number 81600409), grants from The Action Plan for Scientific and Technological Innovation Program from the Shanghai Science and Technology Committee (grant number 19411951500), Shanghai Songjiang District science and technology project (grant number 0702N17002), and Shanghai General Hospital (grant number 06N1702003) to J. Wang, and a grant from the Natural Science Foundation of Shanghai to Y. Shi (grant number 21ZR1448300).

X. Wan, J. Wang, and X. Ye designed, conceived, and supervised the study. X. Wan, J. Wang, and Y. Shi provided funding to support the study. X. Ye performed the experiments. X. Ye and D. Huang collected and analyzed the data. Z. Dong, X. Wang, and J. Xia provided technical support in the *in vivo* experiments. S. Wu, M. Ning, and S. Shen drafted the manuscript. X. Wan and J. Wang revised the manuscript. All the authors approved the final version of the manuscript.

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
