## [Reviewer comments · Microbiology Spectrum]

Microbiology Spectrum

FXR Signaling-Mediated Bile Acid Metabolism is Critical for Alleviation of Cholesterol Gallstones by *Lactobacillus*

Xin Ye, Dan Huang, Zhixia Dong, Xiaoxin Wang, Min Ning, Jie Xia, Shuang Shen, Shan Wu, Yan Shi, Jingjing Wang, and Xinjian Wan

Corresponding Author(s): Xinjian Wan, Shanghai Jiao Tong University Affiliated Sixth People's Hospital, School of Medicine

Review Timeline:

Submission Date:	February 12, 2022
Editorial Decision:	March 19, 2022
Revision Received:	May 19, 2022
Editorial Decision:	June 1, 2022
Revision Received:	July 25, 2022
Editorial Decision:	July 26, 2022
Revision Received:	August 7, 2022
Accepted:	August 10, 2022

Editor: Neha Garg

Reviewer(s): The reviewers have opted to remain anonymous.

Transaction Report:

DOI: <https://doi.org/10.1128/spectrum.00518-22>

March 19, 2022

Prof. Xinjian Wan
Shanghai Jiao Tong University Affiliated Sixth People's Hospital, School of Medicine
Digestive Endoscopic Center
600 Yishan Road ,Shanghai
Shanghai, Shanghai 200030
China

Re: Spectrum00518-22 (FXR Signaling-Mediated Bile Acid Metabolism is Critical for Alleviation of Cholesterol Gallstones by Lactobacillus)

Dear Prof. Xinjian Wan:

Thank you for submitting your manuscript to Microbiology Spectrum. The reviewers have raised a number of concerns, which need to be fully addressed before this manuscript can be accepted. When submitting the revised version of your paper, please provide (1) point-by-point responses to the issues raised by the reviewers as file type "Response to Reviewers," not in your cover letter, and (2) a PDF file that indicates the changes from the original submission (by highlighting or underlining the changes) as file type "Marked Up Manuscript - For Review Only". Please use this link to submit your revised manuscript - we strongly recommend that you submit your paper within the next 60 days or reach out to me. Detailed instructions on submitting your revised paper are below.

Link Not Available

Sincerely,

Neha Garg

Journals Department
Reviewer comments:

Reviewer #1 (Comments for the Author):

This paper focuses on an important issue that must be addressed. The use of probiotics to ameliorate diseases dysbiosis is a very interesting area that must be supported. Therefore, I do think that this paper, with the proper modifications, might be important for the scientific community.

I have some minor and some major concerns, which are listed above:

1. My first question would be: Why these two strains? The authors should indicate the reason for the selection
2. Were the mice siblings? Were they co-housed? This is critical for the experimental design
3. Figure 1G: isn't LD+LP also significantly different from ND?
4. Line_108: cannot understand the link between the two sentences
5. Line 133: why?
6. Line 159: there is no significance marker in the graphic
7. Line 161: weren't the mice supplemented with LR or LP from day one? I don't understand this sentence
8. Line:177 - it is not clear the benefit of the supplementation in the reduction of the gallbladder thickness (figure 1H) - also, comparing the selected images it appears that the wall structure of the gallbladder is completely altered upon LR or LP supplementation.
9. Line184: "empty bubbles"... substitute for vacuoles
10. Figure2: graphics: shouldn't the ND media values be at 1 (in relative amounts) or 100%?
11. Line 262: Figure 2AD should be figure 3AD
12. Line 266-270: I completely disagree with this author's comment, there are no changes in FXR mRNA or protein levels. Thus, in my opinion, they should eliminate this comment.
13. Line 291-293: I do not agree with the authors, there is no difference between ND and LD in Cyp27a1 mRNA or protein.
14. Line295: again, the authors refer to an increase in mRNA levels of CYP7b1, which is not statistically observed.
15. Line306: "insignificantly" should not be used. Concerning this sentence, why is a * in the ND? Is this being compared with what?
16. Line320-324: why were these two markers displayed in the supplementary annex instead of figure 3? In line 331, the authors use these results as a conclusion...
17. Line337: this is a personal choice, but I would substitute "the intestinal flora" with "the gut microbiota"
18. Line 341-343: I don't agree with this comment, there is nothing on the figures that suggest this similarity between ND and supplemented LD mice.
19. Line 345-347: what is the purpose of this analysis? Does this analysis substantially contribute to anything different from the PCoA?
20. Line349-351: where is the statistical data to support this sentence?
21. Line351-353: what? This is not in agreement with figure 4C
22. Figure 4: what do the authors mean by phylum-level? Features?
23. Figure 4D: in the F/B ratio in the y axis should be written "relative F/B ratio", and what is LD+LC?
24. Line 358-359: isn't this to rush to say?
25. Line365: again, where is the statistical analysis to support this comment? Downregulated? Or decreased? Up or downregulation should be used for gene expression not for bacterial community modulation.
26. Line369-371: is it? In my opinion, authors are rushing to connect the gut microbiota to LD.
27. Figure 4F: the p-value compares what groups? NDvsLD?...
28. Line 382-383: If so, why don't these species appear in figure 4G? and why does *L. reuteri* more in the LP than LR supplementation?
29. Line386-387: Again, in what data is this sentence supports?
30. Line 433: comparing the photos the groups with FXR inhibitors seem to have a smaller but increased number of gallstones. Can the authors account for this? Shouldn't the authors perform a control group with both FXR inhibitors without the lactobacillus supplementation?
31. Figure 6: what is aaa, aa, bbb, bb on 4E?

Issues:

Lactobacillus reuteri was renamed *Limosilactobacillus*, this reveals a flaw in authors' update on their work.

Statistics:

The authors used only a t-test to evaluate the statistical differences between the four groups. This is not the preferred test for this type of experiment. ANOVA Tukey for parametric data or ANOVA nonparametric should be the selected statistical analysis. The selection of a T-test for these experiments reduces the robustness of the conclusions.

Also, the authors should state the number of animals used per experiment, also if they removed outliers, please refer to the statistical protocol.

Concerning the number of animals, if they used for instance n=6 in three independent experiments shouldn't be a total of 18 animals per group?

Some graphics do not have the axis information

Figure 4: in A (PCoA), the authors used some groups with 4 and others with 5 animals. Although, in E there are 5 animals per group. My question is, how many animals did the authors use?

Shouldn't the Lactobacillaceae family be highly increased in the supplemented animals? Also, *L. reuteri* is not detected in the LR supplemented mice but was in the LP? How could the authors explain this?

My major issue with this paper is the lack of confidence in the discussion of the results, the authors do not respect the significance values and comment what they "believe" instead of what the data represents!

Venn diagram shows that there are 2 species-specific to LD, isn't this interesting? Why do the authors not comment about it?

Section "Correlation between gut microbiota and host CGS-related parameters": in my opinion authors should rephrase this section, as it is confusing and does not properly inform the reader about what do the authors want with it.

Reviewer #2 (Comments for the Author):

The manuscript by Ye and Huang et al. describes uses a mouse model of gallstones to investigate how two lactobacilli strains interact with the host and the endogenous microbiome to reduce stone formation. The manuscript is a tour de force investigating disease phenotypes, host expression, the bile acid pool, and microbiome composition. The manuscript is a great story; however, it can be difficult to follow at times and I would suggest a number of the main text figure panels could be moved to the supplement to aid in clarity (for example, negative results and/or representative histology images). It could also be helpful if there was a summary figure of the mechanistic findings of the manuscript.

In my opinion the greatest weakness of the manuscript which must be dealt with before publication is in the statistical analysis and microbiome analysis. It appears from the methods that no form of multiple testing correction was applied to most non-microbiome data and that t-tests were used uniformly. It is my suspicion that t-tests would not be appropriate for much of this data (particularly disease outcomes) as I would suspect it is following non-normal distributions. Showing individual points or box plots would greatly help the reader in interpreting the results of the assays and the authors should use appropriate multiple testing corrections and/or non-parametric statistics. In addition gene expression data should almost always be presented and statistically analysed on a log scale (if a t-test is being used).

The analysis of the microbiome data is perhaps the area of the manuscript that requires the most work. The abundances of various taxons appear to be represented and analyzed as read counts which are not relative unless they have been evenly subsampled. This data should be represented as either a proportion/percentage, or probably more accurately as something along the lines of a centered log ratio. While there is not necessarily one correct way to look at differential abundance of 16S rRNA data, there is only objectively one wrong way: t-tests of count data. The authors are encouraged to use a more appropriate method such as *aldex*, *ancom*, or *deseq2*. Correlations against count data are similarly problematic. One more sound method would be to use a transformation such as the centered log ratio to help mitigate some of the issues of compositionally. As it stands the high fraction of microbes with significant correlations is somewhat suspect and hints at an underlying issue in the data analysis. There are also those who would say that more current and higher resolution denoising approaches should be applied to the sequencing data (like *dada2* or *deblur*); however, I do not think this would change the results obtained.

Minor Comments

-The currently accepted nomenclature of both probiotic species have been updated and they should probably be identified by their current names: *Lacticaseibacillus rhamnosus* and *Lactiplantibacillus plantarum*.

-It would be helpful if the statistical tests were identified in the Figure legends.

-The method for bile acid analysis was insufficiently described. How were the bile acids identified/spectated? Were internal standards and/or authentic standards used?

-Figures 2B and 2C are out of order with the text.

-Figure 3B is almost uninterpretable as the signals are so weak. Is this an issue with the image that was put into the manuscript, or was the staining/visualization protocol poor?

-Figure E3: Why are the ND samples not centered at 1?

-Line 264: I am not sure there is any evidence of FXR being differentially expressed from either of these assays irrespective of statistical results. We also would probably not expect to see FXR being differentially regulated would we?

-Figure 4AB: this plot calls into question how the mice were housed and if cage effects may be driving some component of this data. How the mice were caged should be included in the methods.

-For the point the authors are trying to make from their 16S data, it would probably be beneficial to mask the two probiotic strains from the dataset before the distance calculation so they are not contributing to the separation.

-Figure 4B: It does not make sense in any way why this figure would be calculated on the phylum level! Was this a typo?

-Line 354: The F/B ratio is not always associated with obesity and is a somewhat problematic metric when applied to humans. I would remove this statement.

-Line 365: microbes are not down-regulated but rather are present in lower relative abundances.

-Line 399 "At the species level, almost a quarter of these OTUs"- this is a bit of a contradiction, it would appear that you are correlating the species-summarized abundances, not OTUs.

-It is not clear which taxonomic database was used for the 16S data.

-There does not appear to have been metadata deposited along with the sequencing data in the SRA. Some guide to convert from the sample name to the treatment group should be included.

Staff Comments:

Preparing Revision Guidelines

Please return the manuscript within 60 days; if you cannot complete the modification within this time period, please contact me. If you do not wish to modify the manuscript and prefer to submit it to another journal, please notify me of your decision immediately so that the manuscript may be formally withdrawn from consideration by Microbiology Spectrum.

Dear Professor Neha Garg,

Our revised manuscript (ID: Spectrum00518-22.R1) entitled “FXR Signaling-Mediated Bile Acid Metabolism is Critical for Alleviation of Cholesterol Gallstones by Lactobacillus” is re-submitted for publication in the Microbiology Spectrum.

We would like to thank the editor and the reviewers for their instructive and insightful review and for providing us with their comments and suggestions to improve the quality of our manuscript. Based on the critiques from the reviewers, we have provided the point-by-point responses below. We hope that these responses have addressed the concerns. We look forward to your favorable decision.

Thank you very much again.

Yours sincerely,

Wan Xinjian M.D., Ph.D,

Digestive Endoscopic Center,

Shanghai Sixth people's Hospital,

Shanghai Jiao Tong University School of Medicine,

600 Yishan Road,

Shanghai, China 200030,

slwanxinjian2020@126.com.

Reviewer #1

<Major comments >

1. My first question would be: Why these two strains? The authors should indicate the reason for the selection

Thank you for your very helpful suggestion. The reasons for the selection of *Limosilactobacillus reuteri* and *Lactiplantibacillus plantarum* are that these two strains have shown in several studies to lower cholesterol and regulate bile acids (BAs) metabolism. We now stated in the second paragraph of “**Introduction**” (Page 5, Paragraph 2): “Furthermore, it has been reported that *Limosilactobacillus reuteri* CBG-C15 and *Lactiplantibacillus plantarum* CBG-C21 had beneficial effects against hypercholesterolemia in rats¹. Recently, research has shown that *Limosilactobacillus reuteri* Fn041 prevented hypercholesterolemia by promoting cholesterol and bile acids (BAs) excretion in mice². Cong Liang and his colleagues showed that *Lactiplantibacillus plantarum* H-87 has cholesterol-lowering potentials by regulating BA metabolism³. These suggested that bacterial strains belonging to these two *Lactobacillus* species might have the potential to regulate cholesterol related metabolism. We previously isolated *Limosilactobacillus reuteri* CGMCC 17942 (LR) and *Lactiplantibacillus plantarum* CGMCC 14407 (LP) from the feces of healthy adults with normal serum cholesterol level and conducted this study to determine whether these two strains supplementation could reduce gallstones formation and what is the mechanism.”

2. Were the mice siblings? Were they co-housed? This is critical for the experimental design

Thank you for your good advice. In our study, mice were not siblings but they were inbred C57BL/6J mice with the same week age. The mice were not co-housed, and we have stated in “*Mouse strains and treatments*” section of the “**Materials and**

methods” (Page 27, Paragraph 2): “To determine the effect of LR and LP, forty-eight mice were randomly allocated into 4 groups and 12 mice in each group were randomly divided into 4 plastic cages (n=12 per group, n=3 per cage). Next, to further verify the role of FXR, forty-eight mice were randomly allocated into 8 groups. Six mice in each group were randomly divided into 2 plastic cages (n=6 per group, n=3 per cage).”

3. Figure 1G: isn't LD+LP also significantly different from ND?

Thank you for your suggestion. We have used one-way analysis of variance (ANOVA) along with a post-hoc Tukey's test, and found the LD+LP is significantly different from ND ($F(3, 44) = 24.949, P < 0.001$). We now included the data in Figure 1G and stated in “**Results**” (Page 9, Paragraph 1): “The ratio in LP-treated mice did not decrease compared with LD-fed mice and was significantly different from ND group (Figure 1G). The significant increase of the ratio in LP-treated group likely because the body weight loss after LP-treatment (Figure 1G)”, also shown below.

Figure 1

4. Line 108: cannot understand the link between the two sentences

We appreciate the valuable comments and apologies for the confusion. After careful consideration, we think that the sentence from line 108 to line 110 confused the logic

of the paragraph, so we have deleted it.

5. Line 133: why?

We appreciate the helpful critique. We have realized that this description was inappropriate, so we have corrected it to: “We used LD-induced CGS mice model to investigate whether LR and LP could prevent the formation of CGS.” in our revised manuscript (Page 7, Paragraph 3).

6. Line 159: there is no significance marker in the graphic

Thank you for your valuable suggestion. We have added the significance marker in the Figure 1F and shown below (*/# $0.01 < p \leq 0.05$, **/### $0.001 < p \leq 0.01$, ***/### $p \leq 0.001$, * vs ND, # vs LD).

7. Line 161: weren't the mice supplemented with LR or LP from day one? I don't understand this sentence

We appreciate the helpful comment and apologies for the confusion. In our study the mice supplemented with LR or LP were from day one. We have corrected it (Page 8,

Paragraph 2): “The weight gains were inhibited in mice with LR and LP treatments.”

8. Line:177: It is not clear the benefit of the supplementation in the reduction of the gallbladder thickness (figure 1H) also, comparing the selected images it appears that the wall structure of the gallbladder is completely altered upon LR or LP supplementation.

Thank you for your valuable comments. We carefully considered your comments and realized that the description of the reduction of the gallbladder thickness was not accurate. We found that actually the thickness of the tunica adventitia of gallbladder had changed. We have corrected it to: “We have found that the connective tissue of gallbladder’s tunica adventitia was thicker and looser in LD-fed mice compared with that in the ND group, and these alterations were reduced by LR and LP treatments (Figure 1H).” in our revised manuscript (Page 9, Paragraph 2), and marked the tunica adventitia of gallbladder with red arrow in figure 1H shown below.

Figure 1

9. Line184: "empty bubbles"... substitute for vacuoles.

Thank you for your valuable suggestions. We have substituted "empty bubbles" with “vacuoles” in our revised manuscript.

10. Figure2: graphics: shouldn't the ND media values be at 1 (in relative amounts) or 100%?

Thank you for your insightful suggestions. We have made the ND media values of relative amounts be at 1 in figure 2E, G shown below.

Figure 2

11. Line 262: Figure 2AD should be figure 3AD

Thank you for your helpful suggestion and apologies for the mistake. We have corrected it to figure 3A, D in our revised manuscript.

12. Line 266-270: I completely disagree with this author's comment, there are no changes in FXR mRNA or protein levels. Thus, in my opinion, they should eliminate this comment.

Thank you for your valuable suggestions. We have eliminated the comment.

13. Line 291-293: I do not agree with the authors, there is no difference between ND and LD in Cyp27a1 mRNA or protein.

Thank you for your valuable comments. We have corrected it to: “Furthermore, the mRNA and protein levels of CYP27A1, a rate-limiting enzyme in the alternative pathway, were not decreased in LD-fed mice compared with ND-fed mice, and the protein level of CYP27A1 was markedly increased with LR and LP treatments, indicating that the alternative pathway was activated in mice with LR and LP treatments.” in our revised manuscript (Page 13, Paragraph 2).

14. Line295: again, the authors refer to an increase in mRNA levels of CYP7b1, which is not statistically observed.

We appreciate the helpful critique. We have corrected it to: “Then, the expression of CYP7B1 in protein level increased in LD group and reversed with LR treatment (Figure 3G, H).” in our revised manuscript (Page 14, Paragraph 1).

15. Line306: "insignificantly" should not be used. Concerning this sentence, why is a * in the ND? Is this being compared with what?

Thank you for your helpful comments and apologies for the mistakes. We have corrected “insignificantly” to “not statistically significantly” in Page 14, Paragraph 2.

The * in the ND should have been labeled in LD to note the statistical difference between LD and ND ($P < 0.05$). We have corrected it in figure 3I in our revised manuscript.

16. Line320-324: why were these two markers displayed in the supplementary annex instead of figure 3? In line 331, the authors use these results as a conclusion.

We appreciate the valuable comment. We have added the two markers in figure 3, and also shown below.

Figure 3

17. Line337: this is a personal choice, but I would substitute "the intestinal flora" with "the gut microbiota"

Thank you for your helpful suggestion, we have substituted "the intestinal flora" with "the gut microbiota" in our revised manuscript.

18. Line 341-343: I don't agree with this comment, there is nothing on the figures that suggest this similarity between ND and supplemented LD mice.

We appreciate the valuable critique. We have corrected it to: “Principal coordinate analysis (PCoA), a visual method used to study the differences between bacterial communities, indicated that the operational taxonomic units (OTUs) of the LD group were significantly different from those of the ND group, and orally administration of LR and LP shifted the microbial composition of the LD-fed mice toward that of ND group at the OTU level (Bray-Curtis ANOSIM, $R= 0.6668$, $P= 0.001$) (Figure 4A).” in our revised manuscript (Page 16, Paragraph 1).

19. Line 345-347: what is the purpose of this analysis? Does this analysis substantially contribute to anything different from the PCoA?

Thank you for your helpful comments. The rank of distance calculated by analysis of similarities was a further supplementary analysis of the PCoA, which was used to verify the differences between the four groups were larger than the intragroup and to prove the differences in the gut microbiota between groups were caused by different interventions rather than individual differences. Furthermore, in order to reduce the influence of LR and LP on the distance calculation of gut microbiota, we masked the two probiotic strains from the dataset before the distance calculation so they did not contribute to the separation. We have added in our revised manuscript “We masked LR and LP from the dataset for the distance calculation to avoid the two probiotic strains contributing to the separation, and used analysis of similarities (ANOSIM) to compare the similarities of the bacterial communities in the operational taxonomic units (OTUs) level between samples based on Bray-Curtis analysis. Principal coordinate analysis (PCoA), a visual method used to study the differences between bacterial communities, indicated that the OTUs of the LD group were significantly different from those of the ND group, and orally administration of LR and LP shifted

the microbial composition of the LD-fed mice toward that of ND group at the OTU level (Bray-Curtis ANOSIM, $R= 0.6668$, $P= 0.001$) (Figure 4A). The distance between groups was larger than all the intragroup distance, which indicated that the differences between groups were significantly greater than those within the groups (Figure 4B).” in our revised manuscript (Page 15, Paragraph 3).

20. Line349-351: where is the statistical data to support this sentence?

Thank you for your helpful suggestion. The sentence in line349-351 was supported by figure 4C, D. The number in figure 4C showed the average read counts (after subsample) of the gut microbiota of each group. We have added “(Figure 4C, D)” in the end of the sentence.

21. Line351-353: what? This is not in agreement with figure 4C.

We appreciate the valuable critique and apologies for the mistakes. We have corrected it to “The average read counts and the percentage of Bacteroidetes significantly decreased in LD group compared with that in ND group while LP treatment significantly decreased it (Figure 4C, D). Besides, we have found that the read counts of Verrucomicrobia in the LD group decreased from 2128 to 249 compared with that of the ND group, and it markedly increased to 6720 in LP-treated mice (Figure 4C).” in our revised manuscript (Page 16, Paragraph 1).

22. Figure 4: what do the authors mean by phylum-level? Features?

Thank you for your helpful comments. We showed the features of gut microbiota in different groups at the phylum-level, especially Firmicutes, Bacteroidetes, and Verrucomicrobia. The Firmicutes/Bacteroidetes ratio was considered to be closely

related to the body's metabolic functions in several studies^{4,5}, and it has been reported that dietary supplementary of *Lactobacillus salivarius* CML352 would significantly reduce the Firmicutes/Bacteroidetes ratio in hens⁶. Therefore, we focused on the alteration of Firmicutes to Bacteroidetes ratios in LD-fed mice with or without LR or LP treatment. Furthermore, we have found the reduction of Verrucomicrobia in LD-fed mice was reversed by LP-treatment. Meanwhile, we have found the level of *Akkermansia*, one of the most common species of Verrucomicrobia, was markedly increased with LP treatment. Overall, the phylum-level showed the instability of gut microbiota in LD-fed mice and the improvement with LR, LP treatments.

23. Figure 4D: in the F/B ratio in the y axis should be written "relative F/B ratio", and what is LD+LC?

Thank you for your helpful suggestions. We have re-labelled Y axis using "relative F/B ratio" in figure 4D and corrected LD+LC to LD+LP in our revised manuscript and also shown below.

24. Line 358-359: isn't this to rush to say?

Thank you for your valuable suggestions. After careful consideration, we agree that this conclusion was indeed too rush. We have deleted this sentence.

25. Line365: again, where is the statistical analysis to support this comment? Downregulated? Or decreased? Up or downregulation should be used for gene expression not for bacterial community modulation.

Thank you for your valuable suggestions and apologies for these mistakes. The comment was supported by figure 4E, F. We have completed the statistical analysis in figure 4F which would support the comment in our revised manuscript and also shown below. And we have corrected “downregulated” to “decreased”.

Figure 4

26. Line369-371: is it? In my opinion, authors are rushing to connect the gut microbiota to LD.

Thank you for your valuable suggestions. we agree that this connection was indeed too rushing to say. We have deleted this sentence in our revised manuscript.

27. Figure 4F: the p-value compares what groups? NDvsLD?...

Thank you for your helpful comments. The *P*-value and red * in figure 4F showed

statistically significant differences between the groups by Kruskal-Wallis H test. To further compare the alteration of gut microbiota between two of the groups, we have taken the post-hoc (tukey-kramer) analysis to compare each group with LD group and completed the *P*-value in figure 4F in our revised manuscript and show below (* 0.01 < *P* ≤ 0.05, ** 0.001 < *P* ≤ 0.01, *** *P* ≤ 0.001 vs ND, # 0.01 < *P* ≤ 0.05, ## 0.001 < *P* ≤ 0.01, ### *P* ≤ 0.001 vs LD).

Figure 4

28. Line 382-383: If so, why don't these species appear in figure 4G? and why does *L. reuteri* more in the LP than LR supplementation?

Thank you for your helpful comments. Top 50 relative abundances species was included in figure 4G, *L. reuteri* was increased significantly with LR treatment, we now have showed it by a red box below. The abundance of *L. plantarum subsp. plantarum* was increased with LP treatment in supplemental figure 3B, however, it was not in the top 50. When I expanded the scope of the analysis to the top 100, we could find the level of *L. plantarum subsp. plantarum* was increased with LP

treatment and we have showed the community heatmap analysis on Species level (Top 100) below. We have checked that *L. reuteri* did not more in the LP than LR supplementation.

Figure 4
G

Community heatmap analysis on Species level (Top 100)

29. Line386-387: Again, in what data is this sentence supports?

Thank you for your helpful comments. This sentence was supported by figure 4G. It was showed in figure 4G that some species had their own characteristics in the LR and LP groups. We now have added a further analysis of the data “However, LR and LP modulate different in some species. For example, *Pediococcus acidilactic*, *Akkermansia*, *Erysipelotrichacea* and *Erysipelatoclostridium* increased in LP-treated group, not in LR-treated group. *Parasutterella*, *Desulfovibrio sp.* ABHU2SB and *Allobaculum* decreased in LP-treated group, not in LR-treated group. (Figure 4G).” in our revised manuscript (Page 18, Paragraph 1).

30. Line 433: comparing the photos the groups with FXR inhibitors seem to have a smaller but increased number of gallstones. Can the authors account for this? Shouldn't the authors perform a control group with both FXR inhibitors without the lactobacillus supplementation?

Thank you for your valuable suggestions. In our study, we found the groups with FXR inhibitors presented a leaflet or stratified cholesterol crystals instead of round gallstones in LD-fed group. The leaflet or stratified cholesterol crystals was too small to calculate the number precisely. However, in the process of grading gallstones⁷, we made estimates of the number of gallstones and the proportions of gallbladder volume occupied by the gallstones, and found the number of gallstones was increased the groups with FXR inhibitors and most of fine crystals occupy below a half of the gallbladder as the figure 6B showed. It has been reported that FXR gene knockout could promote gallstones formation in mice⁸. In our study, we found FXR signaling pathway was activated by LR and LP treatments, and the groups with both FXR inhibitors with the LR and LP supplementation were sufficient to illustrate that the prevention of gallstones by LR and LP treatments were somewhat dependent on the FXR signaling. We also agree that adding a control group with both FXR inhibitors without the lactobacillus supplementation will make the experiment more perfect, and we will further improve the grouping in future experiments. Thank you again for your

valuable advice.

31. Figure 6: what is aaa, aa, bbb, bb on 4E?

We appreciate the valuable comments and apologies for the confusion. We have added “^a $0.01 < P \leq 0.05$, ^{aa} $0.001 < P \leq 0.01$, ^{aaa} $P \leq 0.001$ vs LD+LR, ^b $0.01 < P \leq 0.05$, ^{bb} $0.001 < P \leq 0.01$, ^{bbb} $P \leq 0.001$ vs LD+LP” in the figure legend of figure 6.

Issues:

Lactobacillus reuteri was renamed Limosilactobacillus, this reveals a flaw in authors' update on their work.

Thank you for your valuable suggestions, we have renamed *Lactobacillus reuteri* as *Limosilactobacillus reuteri* and renamed *Lactobacillus plantarum* as *Lactiplantibacillus plantarum*.

Statistics:

The authors used only a t-test to evaluate the statistical differences between the four groups. This is not the preferred test for this type of experiment. ANOVA Tukey for parametric data or ANOVA nonparametric should be the selected statistical analysis. The selection of a T-test for these experiments reduces the robustness of the conclusions..Also, the authors should state the number of animals used per experiment, also if they removed outliers, please refer to the

statistical protocol. Concerning the number of animals, if they used for instance n=6 in three independent experiments shouldn't be a total of 18 animals per group?

Thank you for your valuable advices. We have changed the T-test to a more appropriate ANOVA along with a post-hoc Tukey's test, and stated in the “*Statistical analysis*” section of the “**Methods**” (Page 33, Paragraph 2): “The obtained data were analyzed statistically to determine the level of significance using one-way analysis of variance (ANOVA) along with a post-hoc Tukey's test. Tukey's post hoc test was performed for data with F at $P < 0.05$ and no significant variance inhomogeneity. Statistical analysis was performed using SPSS 24 (SPSS Inc., Chicago, IL, USA).” We have added the number of animals used per experiment in each figure legend, and we did not exclude any outliers for the analysis in our study. We did not take three independent experiments, and we have deleted the wrong description in the “*Statistical analysis*” section of the “**Methods**”. In the first part of the experiment, we used 12 mice per group, and randomly divided into 4 plastic cages (n=12 per group, n=3 per cage). We randomly selected 6 mice from each group for bile acids analysis and gallstone-related gene and protein expressions detection. And 5 mice were randomly selected from each group for 16S rRNA sequencing analysis. In the first part of the experiment involving FXR inhibitors, we used 6 mice per group, and randomly divided into 2 plastic cages (n=6 per group, n=3 per cage).

Some graphics do not have the axis information

Thank you for your helpful comments. We have added the axis information in our revised manuscript.

Figure 4: in A (PCoA), the authors used some groups with 4 and others with 5

animals. Although, in E there are 5 animals per group. My question is, how many animals did the authors use?

Thank you for your helpful suggestion. We used 5 mice in gut microbiota analysis. In figure 4A, LD3 and LD5 are too close to each other to distinguish, so we marked their sample names for easy identification in the Figure 4A and shown below.

Shouldn't the Lactobacillaceae family be highly increased in the supplemented animals? Also, *L. reuteri* is not detected in the LR supplemented mice but was in the LP? How could the authors explain this?

Thank you for your comment. We have showed the level of Lactobacillaceae family in figure 4F and found the Lactobacillaceae significantly decreased in LD-fed mice. LR and LP treatment both slightly reversed the reduction. It has been reported that the accumulation of cholic acid (CA) could inhibit the growth of *Lactobacillus*.⁹ The reason why the Lactobacillaceae family has not highly increased could be the high concentration of CA (0.5%) in the lithogenic diet. We have checked that *L. reuteri* was detected in the LR supplemented group as showed in supplemental figure 2B and

figure 4G.

My major issue with this paper is the lack of confidence in the discussion of the results, the authors do not respect the significance values and comment what they "believe" instead of what the data represents!

Thank you for your critical comment. We now have more discussions about the information represented by significance values in the "*Discussion*": "The upregulation of the expression of ileal FGF15 and the expression of hepatic SHP in LR/LP-treated mice was caused by the activation of FXR, which downregulated the level of CYP7A1 to inhibit classical pathway for BA synthesis and inhibit the expression of SREBP2, SCAP to reduce cholesterol synthesis. Moreover, it has been reported increased SHP and FGF15 can inhibit cholesterol intestinal absorption." (Page 23, Paragraph 2), "These results suggest that the LR and LP treatments effectively activated FXR signaling to regulate BA and cholesterol metabolism, which could improve the balance between BA and cholesterol." (Page 24, Paragraph 1), "Moreover, we found Akkermansia were significantly positively correlated with FXR signaling pathway related genes (Shp and Fgf15) and the rate-limiting enzyme in the alternative pathway of BA metabolism (Cyp27a1) and BA exportation (Bsep), indicating that Akkermansia in LP-treated mice involved in improving BA metabolism and transportation. The results showed the increase of Akkermansia was several times that of *L. plantarum* in LP-treated mice. A reason for this phenomenon can be explained by which LP produces some metabolites that promote the growth of Akkermansia." (Page 25, Paragraph 2) and "In addition, our study found that *Roseomonas* and *Catabacter* were specific to LD-fed mice and the two strains were always implicated in infection.^{10,11} To our knowledge this is the first time to find the potential relationship between the two strains and CGS." (Page 26, Paragraph 2).

Venn diagram shows that there are 2 species-specific to LD, isn't this interesting? Why do the authors not comment about it?

Thank you for your valuable suggestions. We have added the comments about the two species-specific to LD (*Roseomonas* and *Catabacter*) in the “**Discussion**” (Page 26, Paragraph 2) in our revised manuscript : “In addition, our study found that *Roseomonas* and *Catabacter* were specific to LD-fed mice and the two strains were always implicated in infection.^{10,11} To our knowledge this is the first time to find the potential relationship between the two strains and CGS.” (Page 26, Paragraph 2).

Section "Correlation between gut microbiota and host CGS-related parameters": in my opinion authors should rephrase this section, as it is confusing and does not properly inform the reader about what do the authors want with it.

Thank you for your valuable advice. We have rephrase the section of “**Results**” in our revised manuscript: “To further investigate the correlation between the abundance of different gut microbiota and the factors related to the occurrence of CGS, spearman’s correlation analysis was performed between the top 50 most abundant species and families that were changed by the two strains and several CGS-related parameters, including the incidence and grade of CGS (Figure 5A), blood routine indexes (Figure 5B), expression level of FXR pathway related genes (Figure 5C), and BAs in serum (Figure 5D) and liver (Figure 5C) in all the groups of mice. At the species level, almost a quarter of these species were significantly positively correlated with the incidence and grade of CGS, and one in ten of these species was negatively correlated with them (Figure 5A). Some opportunistic pathogenic bacteria were found to be associated with metabolic disorders in previous studies have also been found to be positively associated with the incidence or grade of CGS, e.g. *Desulfovibrionaceae*, *Desulfovibrio*, *Erysipelatoclostridium*, *Ruminococcaceae* (Figure 5A).^{12,13} We have found that *Akkermansia*,

Muribaculaceae and *Muribaculum* were negatively correlated with both incidence and the grade of CGS (Figure 5A). At the family level, *Akkermansiaceae* and *Defluviitaleaceae* were negatively correlated with both incidence and the grade of CGS, *Lactobacillaceae* and *Muribaculaceae* were negatively correlated with the incidence of CGS (Supplemental figure 4A). Next, we used Spearman's correlation analysis to investigate the relationship between several serum biomarkers and gut microbiota. The results showed that 6 of the 50 species at the species level that were significantly negatively correlated with serum ALP, AST, and ALT, including bacteria belonging to *Akkermansia*, *Muribaculaceae*, and *Muribaculum* (Figure 5B), and 12 of the 50 species at the family level that were negatively correlated with serum ALP, AST, and ALT, namely, *Muribaculaceae*, *Bacteroidales*, *Akkermansiaceae*, *Lactobacillaceae* and so on (Supplemental figure 4B), suggesting that these bacteria might involve in repairing liver injury during the process of CGS formation. We found that some gut microbes were significantly positively correlated with serum TG, TC, HDL, LDL, and Glu, such as *Alloprevotella*, *Dubosiella*, *Roseburia* and *Erysipelatoclostridium* at the species level (Figure 5B). And *Erysipelotrichaceae*, *Family_XIII*, *Atopobiaceae* and so on at the family level were significantly positively correlated with serum TG, TC, HDL, LDL, and Glu too (Supplemental figure 4B), suggesting these bacteria have an impact on metabolic disorders. Furthermore, we analyzed the correlations between gut microbiota and the mRNA levels of liver Cyp7a1, Cyp7b1, Cyp8b1, Fxr, Cyp27a1, Mdr2, Shp, Mrp3, Mrp4, Bsep, Fgfr4 and ileum Fxr, Fgf15, we found *Akkermansia* were significantly positively correlated with FXR signaling pathway related genes (Shp and Fgf15) and Cyp27a1 and Bsep at species and family levels (Figure 5C, Supplemental figure 4C). The abundance of *Enterobacteriaceae* was significantly positively correlated with ileum FXR signaling related gene (ileum Fgf15) and BAs export into bile related genes (Bsep, Mdr2, Mrp3) (Supplemental figure 4C). However, *Muribaculaceae* showed a negative correlation with FXR signaling pathway related genes (Figure 5C, Supplemental figure 4C). These results suggested that the gut microbiota have an impact on the expression profile of genes involved

in FXR signaling pathway and BA synthesis and exportation. Finally, we found that some gut bacteria were significantly correlated with the composition of BAs in serum and liver, suggesting gut microbiota played a vital role in BA metabolism (Figure 5D, E, Supplemental figure 4D, E). These data demonstrate that the CGS phenotype, serum biomarkers, FXR signaling pathway related genes and the composition of BAs were significantly correlated with gut microbiota, and LR, LP treatments exerted protective effects on CGS formation might by regulating the gut microbiota.” (Page 18-20).

Reviewer #2 (Comments for the Author):

The manuscript by Ye and Huang et al. describes uses a mouse model of gallstones to investigate how two lactobacilli strains interact with the host and the endogenous microbiome to reduce stone formation. The manuscript is a tour de force investigating disease phenotypes, host expression, the bile acid pool, and microbiome composition. The manuscript is a great story; however, it can be difficult to follow at times and I would suggest a number of the main text figure panels could be moved to the supplement to aid in clarity (for example, negative results and/or representative histology images). It could also be helpful if there was a summary figure of the mechanistic findings of the manuscript.

Thank you for your valuable comments. Some figures showed negative results and representative histology images of livers in Figure 6 were moved to the supplemental figure in our revised manuscript. And we have made a summary figure of the mechanistic findings of the manuscript as figure 7 and shown below.

Figure 7: Schematic diagram summarizing the mechanisms by which LR and LP prevented CGS formation.

In my opinion the greatest weakness of the manuscript which must be dealt with before publication is in the statistical analysis and microbiome analysis. It appears from the methods that no form of multiple testing correction was applied to most non-microbiome data and that t-tests were used uniformly. It is my suspicion that t-tests would not be appropriate for much of this data (particularly disease outcomes) as I would suspect it is following non-normal distributions. Showing individual points or box plots would greatly help the reader in interpreting the results of the assays and the authors should use appropriate multiple testing corrections and/or non-parametric statistics. In addition, gene expression data should almost always be presented and statistically analysed on a log scale (if a t-test is being used).

We have corrected the T-test to a more appropriate ANOVA along with a post-hoc Tukey's test, and stated in the “*Statistical analysis*” section of the “**Methods**” (Page

33, Paragraph 2): “The obtained non-microbiome data were analyzed statistically to determine the level of significance using one-way analysis of variance (ANOVA) along with a post-hoc Tukey's test. Tukey's post hoc test was performed for data with F at $P < 0.05$ and no significant variance inhomogeneity. Statistical analysis was performed using SPSS 24 (SPSS Inc., Chicago, IL, USA). GraphPad Prism 7.0 (GraphPad Software Inc, La Jolla, CA, USA) was used to shape the experimental data, and the data were presented as the mean \pm SEM (n = 5-12). A P value <0.05 was considered statistically significant. For the gut microbiota data analysis, the abundances of various taxons had been evenly subsampled before analysis and the taxonomy of each OTU representative sequence was analyzed by RDP Classifier version 2.2 against the 16S rRNA database using confidence threshold of 0.7.(68) Principal coordinate analysis (PCoA) based on Bray-Curtis dissimilarity were performed to provide an overview of gut microbial dynamics in response to LD and LR or LP treatments. Similarities of gut microbiota between samples were compared by ANOSIM and Adonis based on Bray-Curtis in phylum level. Statistical comparison of the relative abundance in different groups were analyzed by Kruskal-Wallis H test and using Tukey-kramer test (with 95% confidence intervals, P-value <0.05 , multiple testing P values were adjusted with FDR). The relationship between the top 50 abundance of Species levels and the incidence and grade of CGS and the serum BAs, the serum biochemical values and the CGS-related mRNA expressions were analyzed by Spearman's correlation analysis.” We also have used individual points and box plots to interpret the results of the assays.

The analysis of the microbiome data is perhaps the area of the manuscript that requires the most work. The abundances of various taxons appear to be represented and analyzed as read counts which are not relative unless they have been evenly subsampled. This data should be represented as either a proportion/percentage, or probably more accurately as something along the lines of a centered log ratio. While there is not necessarily one correct way to look at

differential abundance of 16S rRNA data, there is only objectively one wrong way: t-tests of count data. The authors are encouraged to use a more appropriate method such as aldex, ancom, or deseq2. Correlations against count data are similarly problematic. One more sound method would be to use a transformation such as the centered log ratio to help mitigate some of the issues of compositionally. As it stands the high fraction of microbes with significant correlations is somewhat suspect and hints at an underlying issue in the data analysis. There are also those who would say that more current and higher resolution denoising approaches should be applied to the sequencing data (like dada2 or deblur); however, I do not think this would change the results obtained.

The abundances of various taxons all have been evenly subsampled in our study to keep the sample size uniform before analyzing. Therefore, the correlations against read counts were reliable. Then we calculated the relative abundance from the read counts in phylum level and changed the read counts into relative abundance (percentage) in Figure 4D.

Minor Comments

-The currently accepted nomenclature of both probiotic species have been

updated and they should probably be identified by their current names: Lacticaseibacillus rhamnosus and Lactiplantibacillus plantarum.

Thank you for your helpful suggestion. We have corrected *Lactobacillus reuteri* and *Lactobacillus plantarum* to *Limosilactobacillus reuteri* and *Lactiplantibacillus plantarum* in our revised manuscript.

-It would be helpful if the statistical tests were identified in the Figure legends.

Thank you for your helpful suggestion. We have added the statistical tests in the Figure legends in our revised manuscript.

-The method for bile acid analysis was insufficiently described. How were the bile acids identified/spectated? Were internal standards and/or authentic standards used?

Thank you for your good advices. We now stated in the “*Bile acid analysis*” section of the “**Methods**” (Page 30, Paragraph 1): “Accurately weigh 39 kinds of bile acid (BA) standards to prepare a 0.5mM standard solution with methanol. Mix and dilute the 39 kinds of BA standard solutions to 2500, 500, 250, 50, 10, 2.5, 1, 0nM as working standard solutions. The working standard solutions with 8 concentration gradients were determined using LC-MS. Take the concentration of the working standard solution as the abscissa and the peak area as the ordinate, examine the linear range and draw the standard curve. The linear regression equations of each BA were obtained (the correlation coefficient $r > 0.99$) and used for performing quantitative analysis on all samples.”

-Figures 2B and 2C are out of order with the text.

Thank you for your good advices. We have rearranged figures 2B and 2C

-Figure 3B is almost uninterpretable as the signals are so weak. Is this an issue with the image that was put into the manuscript, or was the staining/visualization protocol poor?

Thank you for your helpful suggestion. We have increased the resolution of the image and adjusted the order of pictures to Figure 3A and show below.

-Figure E3: Why are the ND samples not centered at 1?

Thank you for your valuable comment. Since The reason why the ND samples not centered at 1 was that we selected a random value instead of the mean in the ND group for the relative quantitative analysis. We have corrected them in our revised manuscript and shown below.

Figure 3

-Line 264: I am not sure there is any evidence of FXR being differentially expressed from either of these assays irrespective of statistical results. We also would probably not expect to see FXR being differentially regulated would we?

Thank you for your helpful suggestion. We have corrected to “There was no significant difference in the mRNA and protein levels of FXR in ileum and liver between ND and LD groups, and there was still no significant change after LR or LP treatments (Supplemental figure 2A-D).” in our revised manuscript (Page 12, Paragraph 2).

-Figure 4AB: this plot calls into question how the mice were housed and if cage effects may be driving some component of this data. How the mice were caged should be included in the methods.

Thank you for your valuable suggestion. We have stated in “*Mouse strains and treatments*” section of the “**Materials and methods**” (Page 27, Paragraph 2): “To determine the effect of LR and LP, forty-eight mice were randomly allocated into 4 groups and 12 mice in each group were randomly divided into 4 plastic cages (n=12 per group, n=3 per cage).”

-For the point the authors are trying to make from there 16S data, it would probably be beneficial to mask the two probiotic strains from the dataset before the distance calculation so they are not contributing to the separation.

Thank you for your valuable suggestion. We have masked the two probiotic strains from the dataset before the distance calculation in figure 4A, B. And we have corrected to “We masked LR and LP from the dataset for the distance calculation to avoid the two probiotic strains contributing to the separation, and used analysis of similarities (ANOSIM) to compare the similarities of the bacterial communities in the operational taxonomic units (OTUs) level between samples based on Bray-Curtis analysis. Principal coordinate analysis (PCoA), a visual method used to study the differences between bacterial communities, indicated that the OTUs of the LD group were significantly different from those of the ND group, and orally administration of LR and LP shifted the microbial composition of the LD-fed mice toward that of ND group at the OTU level (Bray-Curtis ANOSIM, $R= 0.6668$, $P= 0.001$) (Figure 4A). The distance between groups was larger than all the intragroup distance, which indicated that the differences between groups were significantly greater than those within the groups (Figure 4B).” in our revised manuscript (Page 15, Paragraph 3).

-Figure 4B: It does not make sense in any way why this figure would be calculated on the phylum level! Was this a typo?

Thank you for your valuable suggestion. We apologize for this error sincerely. We have corrected the phylum level to OTU level.

-Line 354: The F/B ratio is not always associated with obesity and is a somewhat problematic metric when applied to humans. I would remove this statement.

Thank you for your helpful suggestion. We have removed the statement in our manuscript.

-Line 365: microbes are not down-regulated but rather are present in lower relative abundances.

Thank you for your helpful advice. We have corrected it to “were present in lower relative abundances” in our revised manuscript (Page 16, Paragraph 2).

-Line 399 "At the species level, almost a quarter of these OTUs"- this is a bit of a contradiction, it would appear that you are correlating the species-summarized abundances, not OTUs.

Thank you for your valuable comments. We apologize for this clerical error sincerely. We have corrected to “species” in our revised manuscript.

-It is not clear which taxonomic database was used for the 16S data.

Thank you for your helpful comment. The SILVA v138 taxonomic database was used for the 16S rRNA data using confidence threshold of 0.7.

-There does not appear to have been metadata deposited along with the sequencing data in the SRA. Some guide to convert from the sample name to the treatment group should be included.

We appreciate the valuable comments and apologies for the confusion. We have updated and corrected the sample name in the SRA

(<https://www.ncbi.nlm.nih.gov/sra/PRJNA680858>).

June 1, 2022

Prof. Xinjian Wan
Shanghai Jiao Tong University Affiliated Sixth People's Hospital, School of Medicine
Digestive Endoscopic Center
600 Yishan Road ,Shanghai
Shanghai, Shanghai 200030
China

Re: Spectrum00518-22R1 (FXR Signaling-Mediated Bile Acid Metabolism is Critical for Alleviation of Cholesterol Gallstones by Lactobacillus)

Dear Prof. Xinjian Wan:

Thank you for submitting your manuscript to Microbiology Spectrum. Both reviewers have raised a few remaining concerns especially with regards to the statistical methods used. These concerns must be addressed before manuscript can be accepted. When submitting the revised version of your paper, please provide (1) point-by-point responses to the issues raised by the reviewers as file type "Response to Reviewers," not in your cover letter, and (2) a PDF file that indicates the changes from the original submission (by highlighting or underlining the changes) as file type "Marked Up Manuscript - For Review Only". Please use this link to submit your revised manuscript - we strongly recommend that you submit your paper within the next 60 days or reach out to me. Detailed instructions on submitting your revised paper are below.

Link Not Available

Sincerely,

Neha Garg

Journals Department
Reviewer comments:

Reviewer #1 (Comments for the Author):

For the authors I just have to add two minor comments:

Line 222-226: it is missing the type of sample total bile acid analysis (serum or liver tissue)

Figure 2: in my opinion authors should have two y-axes for C as the values of TCA completely abolish the view of the other

differences.

Reviewer #2 (Comments for the Author):

The authors have made significant improvements to the manuscript. I have three remaining concerns which were not adequately dealt with, the last requiring significant effort still to address:

Figure 4AB: If there were 12 mice per group (3/cage), why are there only 5 samples per group in the 16S rRNA sequencing data. What happened to the data from the other animals, or how were the animals subsampled for sequencing? This should be clearly described in the methods.

The description of the bile acid analysis is written in a somewhat odd way: more like a protocol than a description of what was done. It is also not clear where the bile acid standards came from, and what validation was done to ensure that they could be adequately resolved in the analytical method.

The authors switched to percentages in the figures; however, defend the use of read counts for correlations. This was and is still problematic. Counts (and percentages) follow a troublesome distribution for both t-tests and most correlations producing unacceptably high false positive rates. This approach is something you will rarely see in robust microbiome research after ~2015. For more information see 10.1038/s41467-022-28034-z, 10.1371/journal.pcbi.1003531, 10.3389/fmicb.2017.02224.

Staff Comments:

Preparing Revision Guidelines

Please return the manuscript within 60 days; if you cannot complete the modification within this time period, please contact me. If you do not wish to modify the manuscript and prefer to submit it to another journal, please notify me of your decision immediately so that the manuscript may be formally withdrawn from consideration by Microbiology Spectrum.

Dear Professor Neha Garg,

Our revised manuscript (ID: Spectrum00518-22.R2) entitled “FXR Signaling-Mediated Bile Acid Metabolism is Critical for Alleviation of Cholesterol Gallstones by Lactobacillus” is re-submitted for publication in the Microbiology Spectrum.

We would like to thank the editor and the reviewers for their instructive and insightful review and for providing us with their comments and suggestions to improve the quality of our manuscript. Based on the critiques from the reviewers, we have provided the point-by-point responses below. We hope that these responses have addressed the concerns. We look forward to your favorable decision.

Thank you very much again.

Yours sincerely,

Wan Xinjian M.D., Ph.D,

Digestive Endoscopic Center,

Shanghai Sixth people's Hospital,

Shanghai Jiao Tong University School of Medicine,

600 Yishan Road,

Shanghai, China 200030,

slwanxinjian2020@126.com.

Reviewer #1 (Comments for the Author):

For the authors I just have to add two minor comments:

Line 222-226: it is missing the type of sample total bile acid analysis (serum or liver tissue)

Thank you for your helpful suggestion and apologies for the mistake. We have corrected it to: “We found that the level of total liver BAs was significantly increased of LD-fed mice compared with that in ND-fed mice, and did not change by LR and LP treatments (Figure 2A). The level of total serum BAs was markedly increased of LD-fed mice compared with ND-fed mice and was significantly decreased by oral LR treatment (Figure 2B).” in our revised manuscript (Page 10, Paragraph 2).

Figure 2: in my opinion authors should have two y-axes for C as the values of TCA completely abolish the view of the other differences.

Thank you for your very helpful suggestion. We have corrected the format of y-axes in Figure 2C, and also shown below.

Figure 2
C

Reviewer #2 (Comments for the Author):

The authors have made significant improvements to the manuscript. I have three remaining concerns which were not adequately dealt with, the last requiring significant effort still to address:

Figure 4AB: If there were 12 mice per group (3/cage), why are there only 5 samples per group in the 16S rRNA sequencing data. What happened to the data from the other animals, or how were the animals subsampled for sequencing? This should be clearly described in the methods.

Thank you for your insightful suggestions. In our study, we only randomly selected 5 mice per group and collected stool samples for 16S rRNA sequencing, the stools of the other mice were frozen in liquid nitrogen. We now stated in the “*DNA extraction*

and bacteria sequencing” section of the “**Methods**” (Page 32, Paragraph 2): “At the end of the experiment, we randomly selected 5 mice from the 12 mice in each group to get stools for gut microbiota detection, the stools of the other mice were frozen in liquid nitrogen.”

The description of the bile acid analysis is written in a somewhat odd way: more like a protocol than a description of what was done. It is also not clear where the bile acid standards came from, and what validation was done to ensure that they could be adequately resolved in the analytical method.

Thank you for your valuable comments. The bile acid standards came from Shanghai Yuanye Bio-Technology Co., Ltd. To validate the bile acid standard solutions were adequately resolved, we selected the appropriate standard curve concentration to make the concentration curve based on the solubility of different standards, as shown in the supplementary table 2 and kept the solutions being clear without any precipitation. We now stated in the “*Bile acid analysis*” section of the “**Methods**” (Page 29, Paragraph 3): “Bile acid (BA) concentrations were determined using negative electrospray liquid chromatography-tandem mass spectrometry (LC-MS/MS). Chromatographic separation was archived on an Acquity UPLC® BEH C18 column (100 mm inner diameter, 1.7 µm; Waters Corp.). Serum and liver tissues were accurately measured, and ethanol (-20 °C) was added for precipitation, and vortex-mixed for 60 seconds then centrifuged. For liver tissues, 100 mg glass beads were added and oscillated before centrifuge. The supernatant was filtered through a 0.22 µm membrane before LC-MS/MS. The mobile phase consisted of a mixture of 0.1% formic acid in water (A) and 0.1% formic acid in acetonitrile (B). Gradient elution was applied for the following phases: 0–4 min, 25% B; 4–9 min, 25–30% B; 9–14 min, 30–36% B; 14–18 min, 36–38% B; 18–24 min, 38–50% B; 24–32 min, 50–75% B; 32–35 min, 75–100% B; 35–38 min, 100–25% B. The flow rate was 0.25 mL/min.

The 40 kinds of BA standards (Yuanye Bio-Technology, Shanghai, China) were prepared in methanol and equally mixed to generate a working standard solution (clear without any precipitation). We used the signal-to-noise ratio (S/N=10) to determine the limit of quantitation (LOQ) of BA standards and selected the appropriate standard curve concentration to make the concentration curve, as shown in the supplementary table 2. Working standard solutions were freshly prepared before using LC-MS/MS. Take the concentration of the working standard solution as the abscissa and the peak area as the ordinate, examine the linear range and draw the standard curve. The linear regression equations of each BA obtained (the correlation coefficient $r > 0.99$) and used for performing quantitative analysis on all samples.”

The authors switched to percentages in the figures; however, defend the use of read counts for correlations. This was and is still problematic. Counts (and percentages) follow a troublesome distribution for both t-tests and most correlations producing unacceptably high false positive rates. This approach is something you will rarely see in robust microbiome research after ~2015. For more information see [10.1038/s41467-022-28034-z](https://doi.org/10.1038/s41467-022-28034-z), [10.1371/journal.pcbi.1003531](https://doi.org/10.1371/journal.pcbi.1003531), [10.3389/fmicb.2017.02224](https://doi.org/10.3389/fmicb.2017.02224).

We appreciate the helpful critique. We have transformed read counts to the centered log ratio for Spearman correlation analysis ($|R| > 0.5$) to reduce the high false positive rates as figure 5 and supplementary figure 5 shown below¹. We now stated in the “*Statistical analysis*” section of the “**Methods**” (Page 34, Paragraph 1): “The relationship between the top 50 abundance of Species levels and the incidence and grade of CGS and the serum BAs, the liver BAs, the serum biochemical indexes and the CGS-related mRNA expressions were analyzed by Spearman’s correlation analysis, and the read counts of the abundances of various taxons had been

transformed to the centered log ratio for Spearman correlation analysis ¹. The Spearman's correlation coefficient $|R| \geq 0.5$ and * $0.01 < P \leq 0.05$, ** $0.001 < P \leq 0.01$, *** $P \leq 0.001$." And we have rephrased the section of "**Results**" in our revised manuscript (Page 18, Paragraph 3): "At the species level, four species and 6 species were significantly positively correlated with the incidence and grade of CGS respectively, and 4 and 2 kinds of these species were negatively correlated with them (Figure 5A). Some opportunistic pathogenic bacteria were found to be associated with metabolic disorders in previous studies have also been found to be positively associated with the incidence or grade of CGS, e.g. *Desulfovibrionaceae*, *Ruminococcaceae* (Figure 5A). ^{2,3} We have found that *Akkermansia*, *Muribaculaceae* and *Muribaculum* were negatively correlated with both incidence and the grade of CGS (Figure 5A). At the family level, *Akkermansiaceae* was negatively correlated with both incidence and the grade of CGS, *Lactobacillaceae* and *Muribaculaceae* were negatively correlated with the incidence of CGS while *Desulfovibrionaceae* were positively correlated with the incidence of CGS (Supplementary figure 4A). Next, we used Spearman's correlation analysis to investigate the relationship between several serum biochemical indexes and gut microbiota. The results showed that 10 of the 50 species at the species level that were significantly negatively correlated with serum ALT, AST, or ALP, including bacteria belonging to *Akkermansia*, *Muribaculaceae*, *Lactobacillaceae*, and *Muribaculum* (Figure 5B), and 10 of the 50 species at the family level that were negatively correlated with serum ALT, AST, or ALP, namely, *Muribaculaceae*, *Akkermansiaceae*, *Lactobacillaceae* and so on (Supplementary figure 4B), suggesting that these bacteria might involve in repairing liver injury during the process of CGS formation. We found that some gut microbes were significantly positively correlated with serum TG, TC, HDL, LDL, and Glu, such as *Desulfovibrio*, *Dubosiella*, *Roseburia* and *Parasutterella* at the species level (Figure 5B). And *Erysipelotrichaceae*, *Defluviitaleaceae*, *Atopobiaceae* and so on at the family level were significantly positively correlated with serum TG, TC, HDL, LDL, and Glu too (Supplementary figure 4B), suggesting these bacteria might have an

impact on metabolic disorders. Furthermore, we analyzed the correlations between gut microbiota and the mRNA levels of liver Cyp7a1, Cyp7b1, Cyp8b1, Cyp27a1, Fxr, Fgfr4, Shp, Mrp3, Mrp4, Mdr2, Bsep and ileum Fxr, Fgf15, we found *Akkermansia* and *Lactobacillus* were significantly positively correlated with liver Fgfr4 at species level (Figure 5C). The abundance of *Enterobacteriaceae* was significantly positively correlated with ileum FXR signaling related gene (ileum Fgf15) and BAs export into bile related genes (Bsep, Mdr2, Mrp3) at species and family levels (Figure 5C, Supplementary figure 4C). However, *Muribaculaceae* showed a negative correlation with liver Shp (Figure 5C, Supplementary figure 4C). These results suggested that the gut microbiota have an impact on the expression profile of genes involved in FXR signaling pathway and BA synthesis and exportation.”

Figure 5

Supplementary Figure 4

Reference

1. Nearing JT, Douglas GM, Hayes MG, MacDonald J, Desai DK, Allward N, Jones CMA, Wright RJ, Dhanani AS, Comeau AM, et al. Microbiome differential abundance methods produce different results across 38 datasets. *Nat Commun* 2022; 13:342.

July 26, 2022

Prof. Xinjian Wan
Shanghai Jiao Tong University Affiliated Sixth People's Hospital, School of Medicine
Digestive Endoscopic Center
600 Yishan Road ,Shanghai
Shanghai, Shanghai 200030
China

Re: Spectrum00518-22R2 (FXR Signaling-Mediated Bile Acid Metabolism is Critical for Alleviation of Cholesterol Gallstones by Lactobacillus)

Dear Prof. Xinjian Wan:

Thank you for submitting your manuscript to Microbiology Spectrum. As you will see your paper is very close to acceptance. Please modify the manuscript along the lines the reviewer has recommended for Table S2. As these revisions are quite minor, I expect that you should be able to turn in the revised paper in less than 30 days, if not sooner. You will find the reviewers' comments below.

When submitting the revised version of your paper, please provide (1) point-by-point responses to the issues I raised in your cover letter, and (2) a PDF file that indicates the changes from the original submission (by highlighting or underlining the changes) as file type "Marked Up Manuscript - For Review Only". Please use this link to submit your revised manuscript. Detailed instructions on submitting your revised paper are below.

Link Not Available

Sincerely,

Neha Garg

Reviewer comments:

Reviewer #2 (Comments for the Author):

One minor concern remains which was not adequately addressed pertaining to the validation of the bile acid method. It is not clear from the authors' response how they validated that each of the 40 bile acids were resolved from each other. Further, because the bile acid nomenclature is so variable, it may be difficult to completely replicate the findings. It would be ideal if Table S2 was updated to include the Supplier #, Cas# and/or inchi key for the analyte, its retention time, and its mass transition.

Preparing Revision Guidelines

- point-by-point responses to the issues I raised in your cover letter
- Upload a compare copy of the manuscript (without figures) as a "Marked-Up Manuscript" file.
- Each figure must be uploaded as a separate file, and any multipanel figures must be assembled into one file.
- Manuscript: A .DOC version of the revised manuscript
- Figures: Editable, high-resolution, individual figure files are required at revision, TIFF or EPS files are preferred

Please return the manuscript within 60 days; if you cannot complete the modification within this time period, please contact me. If you do not wish to modify the manuscript and prefer to submit it to another journal, please notify me of your decision immediately so that the manuscript may be formally withdrawn from consideration by Microbiology Spectrum.

Dear Professor Neha Garg,

Our revised manuscript (ID: Spectrum00518-22.R3) entitled “FXR Signaling-Mediated Bile Acid Metabolism is Critical for Alleviation of Cholesterol Gallstones by *Lactobacillus*” is re-submitted for publication in the *Microbiology Spectrum*.

We would like to thank the editor and the reviewers for their instructive and insightful review and for providing us with their comments and suggestions to improve the quality of our manuscript. Based on the critiques from the reviewer, we have provided the point-by-point responses below. We hope that these responses have addressed the concerns. We look forward to your favorable decision.

Thank you very much again.

Yours sincerely,

Wan Xinjian M.D., Ph.D.,

Digestive Endoscopic Center,

Shanghai Sixth people's Hospital,

Shanghai Jiao Tong University School of Medicine,

600 Yishan Road,

Shanghai, China 200030,

slwanxinjian2020@126.com.

Reviewer comments:

Reviewer #2 (Comments for the Author):

One minor concern remains which was not adequately addressed pertaining to the validation of the bile acid method. It is not clear from the authors' response how they validated that each of the 40 bile acids were resolved from each other. Further, because the bile acid nomenclature is so variable, it may be difficult to completely replicate the findings. It would be ideal if Table S2 was updated to include the Supplier #, Cas# and/or inchi key for the analyte, its retention time, and its mass transition.

Thank you for your valuable suggestions. We have analyzed the bile acids (BAs) working standard solutions by LC-MS/MS, and Total Ion Chromatography (TIC) of working standard solutions was showed in the supplementary figure 6. As the TIC showed that the chromatographic assays of the 40 kinds of BAs were separated well, and the peak shapes were sharp and symmetrical, which validated that each of the 40 BAs were resolved from each other. In addition, we have refined the details of the standards in the Table S3 to present more important information shown below. We now stated in the “*Bile acid analysis*” section of the “**Methods**” (Page 30, Paragraph 2): “The 40 kinds of BA standards (Yuanye Bio-Technology, Shanghai, China) were prepared in methanol and equally mixed to generate a working standard solution (clear without any precipitation). The working standard solutions determined by LC-MS/MS, the Total Ion Chromatography (TIC) of working standard solutions was showed in supplementary figure 6. We used the signal-to-noise ratio (S/N=10) to determined limit of quantitation (LOQ) of BA standards and selected the appropriate standard curve concentration to make the concentration curve, as shown in the supplementary table 2. The Supplier #, Cas# and InChI Key for the standard analytes and their retention times, and the mass transition ion pairs (m/z) (Precursor Ion and Product Ion) were presented in the supplementary table 3. Working standard solutions were freshly prepared before using LC-MS/MS. Take the concentration of the working standard solution as the abscissa and the peak area as the ordinate, examine the linear range and draw the

standard curve. The linear regression equations of each BA obtained (the correlation coefficient $r > 0.99$) and used for performing quantitative analysis on all samples.”

Supplementary Figure 6 The Total Ion Chromatography (TIC) of Working Standard Solutions.

Supplementary Table 3 Detailed Information of Each Bile Acid

Name	Full Name	CAS #	Supplier #	InChI Key	Retention Time (min)	Precursor Ion	Product Ion
alloLCA	Allolithocholic acid	2276-93-9	B27763	SMEROWZSTRWXGI-NWFSOSCSSA-N	29.92	375.145	375.145
LCA	Lithocholic acid	434-13-9	B28100	SMEROWZSTRWXGI-HVATVPOCSA-N	31.92	375.3	375.3
isoLCA	Isolithocholic acid	1534-35-6	B65766	SMEROWZSTRWXGI-WFVDQZAMSA-N	30.32	375.301	375.301

NorDCA	23-Nordeoxycholic acid	53608-86-9	B74559	PLRQOCVIINWCFA-AHFDLSHQSA-N	24.36	377.3	377.3
6-ketoLCA	6-ketolithocholic acid	106439-47-8	B79566	JWZBXXKZZDYMDCJ-IJPFKRJSSA-N	28.99	389.238	389.238
12-ketoLCA	12-ketolithocholic acid	5130-29-0	S22145	CVNYHSDFZXHMMJ-VPUMZWJWSA-N	24.42	389.301	389.301
7-ketoLCA	7-ketolithocholic acid	4651-67-6	B26465	DXOCDBGWDZAYRQ-AURDAFMXSA-N	23.79	389.302	389.302
β -UDCA	3 β -Ursodeoxycholic acid	78919-26-3	B72964	RUDATBOHQWOJDD-YAKZLEHISA-N	19.65	391.254	391.254
DCA	Deoxycholic acid	83-44-3	B21032	KXGVEGMKQFWNSR-LLQZFEROSA-N	27.32	391.3	391.3
CDCA	Chenodeoxycholic acid	474-25-9	B20347	RUDATBOHQWOJDD-BSWAIDMHSA-N	26.69	391.301	391.301
UDCA	Ursodeoxycholic acid	128-13-2	B21405	RUDATBOHQWOJDD-UZVSRGJWSA-N	21.43	391.302	391.302
HDCA	Hyodeoxycholic acid	83-49-8	B21672	DGABKXLVXPYZII-SIBKNCMHSA-N	21.83	391.303	391.303
NorCA	Norcholic acid	60696-62-0	B74560	SHUYNJFEXPRUGR-RTCCEZQESA-N	15.95	393.211	329.1
DHCA	Dehydrocholic acid	81-23-2	B24725	OHXPGWPVLPUSM-KLRNGDH RSA-N	13.78	401.2	401.2
7,12-diketo LCA	7,12-diketolithocholic acid	517-33-9	B27766	MAFJMPFLJJCSTB-FQBQTYDJSA-N	13.16	403.14	403.14
6,7-diketo LCA	6,7-diketolithocholic acid	N/A	S40141	FRIRHJVKXFYECW-ZTERCDMUSA-N	23.95	403.3	403.3

α -MCA	α -Muricholic acid	2393-58-0	S22142	DKPMWHFRUGMUKF-GDYCBZMLSA-N	15.68	407.3	407.3
UCA	Ursocholic acid	2955-27-3	B22360	BHQCQFFYRZLCQQ-UTLSPDKDSA-N	12.64	407.301	407.301
β -MCA	β -Muricholic acid	2393-59-1	B74541	DKPMWHFRUGMUKF-CRKPLTDNSA-N	16.46	407.302	407.302
CA	Cholic acid	81-25-4	B20274	BHQCQFFYRZLCQQ-OELDTZBJSAN	20.81	407.303	407.303
ACA	Allocholic acid	2464-18-8	B22337	BHQCQFFYRZLCQQ-PGHAKIONSA-N	20.33	407.304	407.304
β CA	3 β -Cholic acid	3338-16-7	T51018	BHQCQFFYRZLCQQ-UXWVVXDJSA-N	15.13	407.362	407.362
GLCA	Glycolithocholic acid Sodium Salt	24404-83-9	B74545	LQKBJAKZKFBLIB-LGURPPGFSA-M	27.83	432.401	73.9
GHDCA	Glycohyodeoxycholic acid	13042-33-6	B27133	SPOIYSFQOFYOFZ-BRDORRHWSA-N	15.28	448.2	74
GCDCA	Glycochenodeoxycholic acid Sodium Salt	16564-43-5	S31335	AAAYACJGHNRIFFT-YRJJIGPTSA-M	21.59	448.276	73.9
GUDCA	Glycoursodeoxycholic acid	64480-66-6	B32965	GHCZAUBVMUEKKP-XROMFQGDSA-N	15	448.277	73.9
GDCA	Glycodeoxycholic acid Sodium Salt	16409-34-0	B27134	VMSNAUA EKXEYGP-YEUHZSMFSA-M	22.63	448.279	73.9
LCA-3S	Lithocholic acid 3-sulfate	64936-81-8	T70708	AXDXVEYHEODSPN-HVATVPOCSA-N	26.59	455.196	96.9
GCA	Sodium Glycocholate Hydrate	338950-81-5	S31334	YWROUPEFMHKARON-HJRQWJHVSA-M	15.27	464.281	73.9

TLCA	Taurolithocholic acid Sodium Salt	6042- 32-6	S26225	YAERYJYXPRIDTO- HRHHVWJRSA-M	24.72	482.223	80
THDCA	Taurohyodeoxycholic Acid	110026- 03-4	B27364	HMXPOCDLAFANT- BHYUGXBJSAN	12.22	498.224	79.9
TUDCA	Tauroursodeoxycholic Acid	14605- 22-2	B20921	BHTRKEVKTCKXOH- LBSADWJPSAN	12.22	498.25	80
TDCA	Taurodeoxycholic acid Sodium Salt	1180- 95-6	S30708	YXHRQQJFKOHLAP- FVCKGWAHSA-M	19.1	498.35	79.8
TCDCA	Taurochenodeoxycholic acid	516-35- 8	B20919	BHTRKEVKTCKXOH- BJLOMENOSAN	17.68	498.357	79.8
TCA	Taurocholic acid Sodium Salt	145-42- 6	B20918	JAJWGJBVLPIOOH- QGRZANFFSAM	12.9	514.332	79.8
T- α - MCA	Tauro- α -muricholic acid	25613- 05-2	B74540	XSOLDPYUICCHJX- UHFFFAOYSAN	7.41	514.337	79.9
THCA	Taurohyocholic acid	32747- 07-2	YY80220	XSOLDPYUICCHJX- QZEPYOAJSAN	10.25	514.343	79.9
T- β - MCA	Tauro- β -muricholic acid	25696- 60-0	B27767	XSOLDPYUICCHJX- OEYGYFRSSAN	7.74	514.346	79.9
CDCA- G	Chenodeoxycholic acid 24-Acyl- β -D- glucuronide	208038- 27-1	T48407	ZTJBLIAPAIPNJE- BWGRGVIUSAN	20.33	567.525	391.1
GUCA	Glycoursocholic acid	95093- 95-1	YY91138	RFDAIACWWDREDC- IFZPJRIXSAN	36.74	416.2	73.9

August 10, 2022

Prof. Xinjian Wan
Shanghai Jiao Tong University Affiliated Sixth People's Hospital, School of Medicine
Digestive Endoscopic Center
600 Yishan Road ,Shanghai
Shanghai, Shanghai 200030
China

Re: Spectrum00518-22R3 (FXR Signaling-Mediated Bile Acid Metabolism is Critical for Alleviation of Cholesterol Gallstones by Lactobacillus)

Dear Prof. Xinjian Wan:

Your manuscript has been accepted, and I am forwarding it to the ASM Journals Department for publication. You will be notified when your proofs are ready to be viewed.

Sincerely,

Neha Garg
Editor, Microbiology Spectrum

Journals Department
Supplemental Information: Accept